# GENERATION SPACE SIZE: UNDERSTANDING AND CALIBRATING OPEN-ENDEDNESS OF LLM GENERATIONS

## ABSTRACT

Different open-ended generation tasks require different degrees of output diversity. However, current LLMs are often miscalibrated. They collapse to overly homogeneous outputs for creative tasks and hallucinate diverse but incorrect responses for factual tasks. We argue that these two failure modes are unified by, and can both be addressed by, the notion of *effective generation space size* (GSS) — the set of semantically distinct outputs a model considers for a prompt. We present GSSBench, a task suite of prompt pairs with ground-truth GSS relationships to assess different metrics and understand where models diverge from desired behavior. We find that hallucination detection metrics, particularly EigenScore, consistently outperform standard diversity and uncertainty quantification metrics, providing interpretable insights into a model's internal task representations for the open-endedness of different prompts. We demonstrate three applications of GSS: (1) detecting prompt ambiguity and when models ask clarification questions for better grounding, (2) interpreting overthinking and underthinking in reasoning models, and (3) steering models to expand their generation space to yield high-quality and diverse outputs.

## 1 INTRODUCTION

When a person answers a question, the breadth of possibilities they consider depends on the task at hand. For example, when brainstorming with a collaborator, one may cast a wide net, exploring far-flung possibilities in search of creative connections. On the other hand, a trivia question requires narrowing one's focus to retrieve specific, accurate information. As it is challenging to systematically articulate the full space of "what comes to mind" (Mills & Phillips, 2023; Phillips et al., 2019; Bear et al., 2020) for a query, researchers rely on produced speech or text as proxies. Similarly, for large language models (LLMs), though we can infer the generation space size from outputs, we cannot directly access what the model implicitly "considers" – what we call its *effective generation space*.

Prior work has identified two failure modes that we relate to generation space size (GSS). First, on creative tasks where diversity is desired, models produce overly homogeneous outputs, with post-training causing further collapse (West & Potts, 2025; Moon et al., 2024; Kirk et al., 2023; Li et al., 2024). Second, on constrained tasks where accuracy matters, models hallucinate, their generation space expanding beyond correct answers (Nikitin et al., 2024; Farquhar et al., 2024; Kuhn et al., 2023). Typical approaches have tried to address these problems separately: either maximizing diversity signals (Lanchantin et al., 2025; Li et al., 2025) or constraining it for factual accuracy (Huang et al., 2024; Vashurin et al., 2024; Detommaso et al., 2024; Zhao et al., 2024; Shi et al., 2025; Liu et al., 2025). We unify these as two sides of the same problem: GSS miscalibration.

To measure and understand GSS miscalibration, we need a systematic way to evaluate how well different metrics serve as proxies for a model's generation space. To address these gaps, we propose **GSSBench**, an evaluation framework using prompt pairs with known GSS relationships (e.g., "Write an email to Dan" has a smaller GSS than "Write an email"). This framework enables us to both (1) identify which metrics best approximate a given model's GSS and (2) determine which models are better calibrated under a given metric. We find that hallucination detection metrics, particularly EigenScore (Chen et al., 2024), best approximate GSS across all models tested, and that scaling does not necessarily improve GSS calibration.

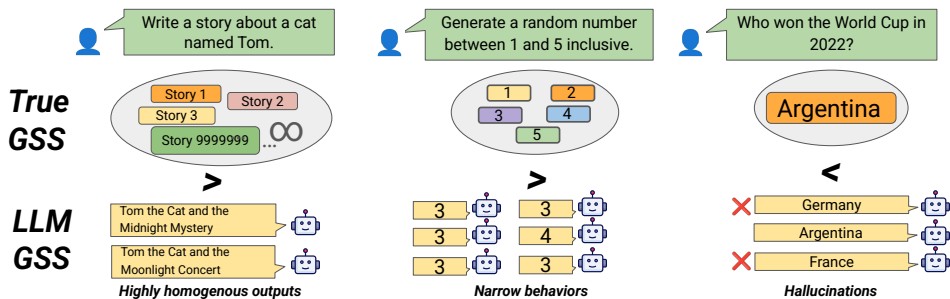

Figure 1: Overview of two failure modes of LLMs under the lens of generation space size. An LLM may generate overly homogenous responses when the true GSS ought to be larger (left) or generate incorrect hallucinations when the true GSS ought to be small (right).

**Contributions** Our contributions are: (1) the formalization of GSS as a unifying framework for understanding various model failures, such as output homogeneity and hallucination (Figure 1); (2) GSSBench, an evaluation suite for measuring GSS and its miscalibration; and (3) case studies of the utility of GSS measurement for grounding, reasoning analysis, and diversity optimization.

## 2 MEASURING GENERATION SPACE SIZE

### 2.1 PRELIMINARIES

For every prompt $p$, there is a ground truth generation space $G_t(p)$: the semantic distribution of all possible correct outputs. This space can range from very small (e.g. for factual QA with one correct answer) to infinitely large (e.g. for open-ended creative tasks). While it can be difficult to quantify the concrete $G_t$ for open-ended tasks, we know that some spaces are larger than others, e.g., the space of "Generate an email that contains the word *Sam*" is smaller than the space for the prompt "Generate an email." A model $m$ also has a generation space $G_m(p)$: the space of outputs that a model "considers", i.e., could generate for a given prompt. We interpret previous work on LLMs' failure modes as the misalignment between a model's generation space $G_m(p)$ and the desired generation space $G_t(p)$: the model's GSS $|G_m(p)|$ may either be smaller or larger than the desired GSS $|G_t(p)|$ (where $|G|$ is the size of the generation space $G$). For a given prompt, a model's GSS is:

$$|G_m(p)| = |G_t(p)| + \varepsilon_m(p) \tag{1}$$

That is, there is some error $\varepsilon_m$ between the model's GSS and the desired GSS.

Moreover, it is currently impossible to access the model's generation space $G_m$. But if we can obtain a more direct proxy for GSS, then we can more feasibly understand model behaviors and calibrate the model's generation space to the true generation space. Thus, we aim to find a mapping function $f_m(p)$ from a prompt $p$ and a model $m$ as a proxy measure of the GSS $|G_m(p)|$. We hypothesize that concepts such as uncertainty quantification, diversity measurements, and hallucination detection are closely related to GSS, and thus use related metrics as candidates for $f$. Each such metric is an imperfect proxy, i.e.,

$$|G_m(p)| = f_m(p) + \delta_{f,m}(p), \tag{2}$$

where $f_m(p)$ is the metric score (e.g., entropy) for the given prompt $p$ and $\delta_{f,m}(p)$ denotes the error between the metric score and the real $G_m(p)$. Our key insight is as follows: on prompts where we

| Dataset | Prompt A | Prompt B |
|---|---|---|
| Complement | Generate a poem about the moon. | Generate anything that is not a poem about the moon. |
| FactualQA | What is the fastest land animal? | Name a land animal. |
| Random Choice | Choose one from the following: cyan, pink. | Choose one from the following: red, orange, pink, cyan, purple, black |
| Subset | Write a Python program for converting CSV to JSON | Write a Python program. |
| Union | Come up with an idea for a song. | Come up with an idea for a song or a poem or a movie or a book. |
| Intersection | Write a poem using rhyming couplets, limited to 8 lines. | Please write a poem. |

Table 1: **GSSBench Datasets.** We construct datasets such that prompt A has smaller GSS than prompt B. Note that generation size and prompt length are not correlated (more in Appendix A.5).

know the ground truth desired GSS $|G_t|$, we can (1) **find metric $f_m$ that best approximates a model's GSS**, i.e.,

$$\arg\min_f \mid \delta_{f,m}(p) \mid = \arg\min_f \mid f_m - |G_m| \mid \approx \arg\min_f \mid f_m - |G_t| \mid . \tag{3}$$

That is, by assuming that $|G_m(p)| \approx |G_t(p)|$, i.e., $|\varepsilon_m(p)|$ is sufficiently small that this has signal (we validate in Appendix A.2 that $|\varepsilon_m(p)|$ is indeed very small and does not impact model orderings on the Random Choice dataset, using the number of unique generations as a direct measurement of a model's generation space), we measure which metric $f$ is closest to the ground truth $G_t$ and thus also to the model's GSS $G_m$.

2) We are also interested in measuring how calibrated a model's generation space size is, i.e., **comparing models to see which model's GSS is closest to the desired ground truth**, i.e., minimizing the miscalibration error $\varepsilon_m = ||G_m| - |G_t||$. Again, since we don't have access to $|G_m|$, but can identify a metric $f$ that approximates it as $f_m \approx |G_m| + \delta_{f,m}$, our minimization problem becomes:

$$\arg\min_M \mid f_m + \delta_{f,m} - |G_t| \mid \approx \arg\min_m \mid f_m - |G_t| \mid , \tag{4}$$

where we similarly assume that $|\delta_{f,m}|$ is sufficiently small for a good proxy $f_m$.

Thus, given prompts where we know the ground truth desired GSS $|G_t|$ (which we provide with our evaluation framework GSSBench in the next section), we can (1) find metric $f_m$ that best approximates a particular model $m$'s GSS and (2) compare across models to understand which models' GSS is closest to the ground truth or are otherwise miscalibrated.

## 2.2 GSSBench: A Bidirectional Evaluation Framwork

**Datasets**   As it is often hard to quantify the desired ground truth GSS for a prompt — particularly for open-ended tasks — we use set-theoretic operations to create pairs of prompts, $\langle x, y \rangle$, where the set-theoretic relationship between $x$ and $y$ yields a clear comparison in terms of GSS, such that $G_t(x) > G_t(y)$. With this set-up, we construct the following six synthetic datasets, resulting in 9300 prompt pairs $(x, y)$ where $|G_t(x)| > |G_t(y)|$ (examples in Table 1). For each prompt pair, we used GPT-4o to determine the prompt with the bigger GSS and reached high agreement.

The prompt pairs include: (1) **Complement**: We take the complement of a prompt like "Generate a poem about the moon" to be "Generate *anything that is not* a poem about the moon". The latter has a much larger generation space. We generate 500 pairs of base prompts of open-ended generation tasks (e.g. email generation, persona generation, etc.) plus complement versions for each. (2) **factualQA**: We create a synthetic dataset of 500 prompt pairs of FactualQA questions where one generation task comes with a wider range of correct candidate answers (such as "Name a river" versus "Name a river in Brazil"). (3) **Random Choice**: We can explicitly enumerate a set $S$ in the prompt and instruct the model to pick an item from $S$. By varying the size of $S$ across prompts, we can more directly control the possible generations to choose from. The number of unique generations across samples can be used to validate the true size of the space. (4) **Subset**: We create a generic generation task (e.g. email, Python script, persona, poem, or short story) and keep appending additional requirements at the end, resulting in 5 prompts of varying levels of specificity (and 10 pairs for comparison) in each set and a total of 180 sets. (5) **Union**: For each set, we create 4 base prompts (e.g. come up with an idea for breakfast/lunch/dinner/afternoon snack), then take the union of each subset, resulting in 15 prompts per set (50 comparisons in each set). We created a total of 60 such sets. (6) **Intersection**: Similar to the Union dataset, we first create 4 base prompts for each set (e.g. write an email, write 200 words, write 3 paragraphs, and write in formal language) and include 60 sets in total. For each set, we take the intersections of the base prompts, resulting in 3000 comparisons in total (full details in A.1).

**Evaluation criteria**   For each model-metric pair $(m, f)$, we evaluate a given function $f$'s alignment between the predicted ordering of generation space sizes and the ground-truth ordering using pairwise accuracy $\text{Acc}(m, f)$ for each prompt pair, where the model-metric pair receives a score of 1 if $f(x) > f(y)$ (where $G_t(x) > G_t(y)$) and 0 otherwise. This enables us to identify:

$$f^\star(m) = \arg\max_{f \in \mathcal{F}} \text{Acc}_m(f), \tag{5}$$

i.e. a metric $f$ that maximizes a given model's accuracy on our task, thus minimizing the error $\delta_{f,m}$ and serving as the best proxy for this model's GSS (corresponding to Equation 3).

We are also interested in measuring the miscalibration of models' GSS to **identify the model whose GSS is closest to the ground truth**, conditioned on the metric. That is, for a set of models $M$, we are interested in finding the model $m$ that achieves the highest accuracy (corresponding to Equation 4). With $f_m$ approximating $|G_m|$, we can compare $m \in M$ conditioned on the metric $f$ to identify:

$$m^\star(f) \;=\; \arg\max_{m \in \mathcal{M}} \; \mathrm{Acc}_m(f). \tag{6}$$

**Mapping function candidates** We evaluate the following metrics as candidates for $f$: perplexity (Shannon, 1951), energy (Liu et al., 2020), length-normalized entropy (Malinin & Gales, 2020), lexical similarity (Lin et al., 2023), EigenScore and its two variants (Chen et al., 2024), and semantic entropy (Kuhn et al., 2023; Farquhar et al., 2024). **Perplexity** and **length-normalized entropy** have long been used in uncertainty quantification. **Energy** is an OOD detection method that reflects whether a prompt aligns with the model's learned distribution. **Lexical similarity** captures the semantic similarities of sampled outputs and operates at the output level. **Semantic entropy** is an effective tool for hallucination detection that calculates the log likelihoods of each sampled generation, clusters them based on entailment relationships, and aggregates probabilities across semantically similar clusters. **EigenScore**, also originally proposed for hallucination detection (Chen et al., 2024), is computed by constructing a covariance matrix of the sentence embeddings of $K$ samples and computing its logarithm determinant. We explore a variant of the original implementation of EigenScore used in Chen et al. (2024) $E_{\mathrm{average}}$, which averages across layers and tokens (the original implementation takes the last hidden layer and the last embedding). As an additional ablation, we introduce $E_{\mathrm{output}}$, which obtains sentence embeddings from an external sentence embedding model (`Roberta Large V1`), representing differential entropy in the embedding space. For all metrics, we perform ablation studies on differerent model parameters (more details in Appendix B) and set the final temperature in our experiments to 1, sample size to 10, and top-k to 10 based on ablation results.

**Models** We evaluate the following five models: Llama-8B-Instruct (Dubey et al., 2024), Mistral-7B-v0.3 (Jiang et al., 2023), Qwen3-0.6B (Yang et al., 2025), Qwen3-4B (Yang et al., 2025), and Qwen3-8B (Yang et al., 2025). We choose all instruction-tuned models to ensure that the models can respond appropriately to open-ended tasks so that the miscalibration error is relatively smaller than non-instruction-tuned models. We experiment with relatively smaller models for computational efficiency and use the three model sizes of Qwen-3 to examine the effects of scaling.

## 3 GSSBENCH RESULTS

**EigenScore variants are the best-performing metrics** For each response, we used GPT-4o to rate its validity and only include instances of high-quality responses in our analysis. We find that the two versions of EigenScore — $E_{\mathrm{output}}$ and $E_{\mathrm{average}}$ — achieve the highest accuracy across the five models, outperforming other metrics like perplexity and lexical similarity (Table 2). This consistently higher performance suggests that EigenScore is a good proxy for a model's GSS. We further see that $E_{\mathrm{output}}$ and $E_{\mathrm{average}}$ have bimodal distributions, which corresponds to these metrics meaningfully separating between prompts with smaller versus larger GSS, while the distributions are more overlapping for other metrics (Figure 2).

Table 2: **GSSBench performance across models and metrics.** We show the average accuracy on GSSBench for each metric for each model (with each of the six datasets weighted equally, excluding responses that are considered low-quality). The best-performing metric for each model is **bolded**, and the best-performing model for each metric is *italicized*.

| Model | Perplexity ↑ | Energy ↑ | Entropy ↑ | Lex Sim ↓ | $E_{\mathrm{output}}$ ↑ | $E_{\mathrm{average}}$ ↑ | Sem En ↑ |
|---|---|---|---|---|---|---|---|
| Llama-8B-Instruct | *0.571* | *0.586* | *0.621* | *0.659* | **0.720** | 0.705 | 0.534 |
| Mistral-7B | 0.360 | 0.576 | 0.454 | 0.554 | 0.621 | **0.715** | 0.478 |
| Qwen3-0.6B | 0.492 | 0.500 | 0.439 | 0.590 | ***0.761*** | 0.672 | *0.563* |
| Qwen3-4B | 0.491 | 0.534 | 0.506 | 0.535 | **0.604** | 0.589 | 0.456 |
| Qwen3-8B | 0.444 | 0.388 | 0.448 | 0.483 | 0.583 | **0.621** | 0.445 |

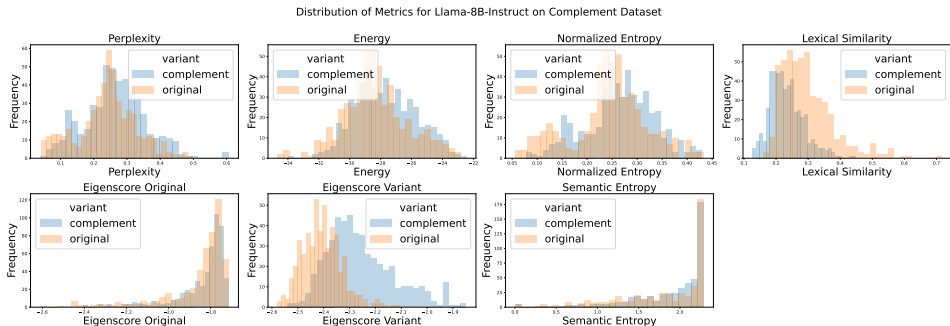

Figure 2: The distribution of metric scores for prompts with smaller GSS (original) versus larger (complement). Here we show the distributions for Llama-8B-Instruct on the Complement Dataset as an example; see Appendix A.7 for all models and datasets. Examples of prompts in the overlapping area are in Tab A17.

**Llama-8B and Qwen-0.6B have highest accuracy**    GSSBench enables the comparison of models' calibration for a given metric. We find that Llama-8B-Instruct is the most well-calibrated model for most metrics except for $E_{\text{output}}$ and semantic entropy, where Qwen3-0.6B has higher accuracy. Comparison across the three model sizes of Qwen3 (0.6B, 4B, and 8B) shows that larger models are not necessarily better calibrated: 0.6B outperforms 8B on all metrics. This corroborates prior work finding that larger instruction-tuned models perform worse on random generation tasks (West & Potts, 2025), a finding that may in part explain why the larger model we test also has lower accuracy overall. Finally, GSSBench enables the analysis of behaviors on different tasks for the same model, revealing specific calibration failures: Llama-8B-Instruct, for example, is well-calibrated on Complement but struggles on Random Choice; Qwen3-4B, on the other hand, is well-calibrated on Random Choice but not on factualQA (see Tab A16 for results by datasets).

## 4    APPLICATIONS OF GSS MEASUREMENT AND CALIBRATION

Our concept and quantification of GSS can unify three previously-separate failures — across the domains of human-LLM interaction, reasoning, and fine-tuning — as miscalibrations of GSS. First, we show that LLMs' failure to perform conversational grounding, i.e., respond appropriately by asking for clarification when users pose ambiguous queries, can be viewed and measured as a miscalibration of GSS. Second, GSS provides insights into the space of possible generations for reasoning models and when they might "underthink" or "overthink" problems. Third, GSS can be used to address the mode collapse that can occur in preference alignment: we show that using GSS proxies in the reward function results in comparable performance with previous approaches that rely on post-hoc diversity metrics (Lanchantin et al., 2025; Li et al., 2025). For each of these tasks, we demonstrate that EigenScore in particular — the best proxy that we identify for GSS — similarly has the highest performance on each of these tasks compared to other metrics.

### 4.1    USING GSS TO MEASURE PROMPT AMBIGUITY AND ASKING FOR CLARIFICATION

On ambiguous prompts, LLMs exhibit undesired behaviors of making assumptions rather than asking clarifying questions (Shaikh et al., 2023; 2025). Here we show that GSS can help diagnose and potentially address this behavior.

**Experiment 1: GSS measures prompt ambiguity**    Shaikh et al. (2025) introduce **RIFTS**, a dataset of 1740 prompts distinguishing between *ambiguous* prompts that require clarification with the user versus *non-ambiguous* ones that do not require clarification. We examine whether different metrics can separate between the ambiguous and non-ambiguous prompts in RIFTS. To test the hypothesis that ambiguous prompts correspond to larger GSS in a model's representation, we perform a two-sample Welch's $t$-test to examine whether the mean of the ambiguous prompts are significantly higher than the mean of the non-ambiguous prompts. We found that only $E_{\text{output}}$ and $E_{\text{average}}$ correctly

separate the two classes for most models. In particular, $E_{\text{output}}$ can correctly separate the two classes for every model tested (Table 3).

Table 3: **Different metrics' ability to separate ambiguous vs. non-ambiguous prompts on RIFTS across models (top) and prompts that lead to clarification questions vs. those that do not (bottom).** Values are $t$-statistics of whether the two sets of prompts have significantly different means. Higher is better for all metrics except lexical similarity. Stars denote significance ($^*p < 0.05$, $^{**}p < 0.01$, $^{***}p < 0.001$, (ns) not significant). Significant values (in the correct direction) are in **green**.

| Task | Model | Perplexity ↑ | Energy ↑ | Entropy | Lex Sim ↓ | $E_{\text{output}}$ ↑ | $E_{\text{average}}$ ↑ | Sem En ↑ |
|---|---|---|---|---|---|---|---|---|
| RIFTS | Llama-8B-Instruct | 0.24 (ns) | **2.09**$^*$ | 0.61 (ns) | -1.27 (ns) | **5.47**$^{***}$ | **5.17**$^{***}$ | **2.41**$^*$ |
| | Mistral-7B | -1.78 (ns) | 0.13 (ns) | -3.64$^{***}$ | -0.99 (ns) | **2.74**$^{**}$ | -1.20 (ns) | 1.46 (ns) |
| | Qwen3-0.6B | -2.14$^*$ | -0.96 (ns) | -2.99$^{**}$ | 0.34 (ns) | **6.47**$^{***}$ | 0.93 (ns) | **3.06**$^{**}$ |
| | Qwen3-4B | -3.82$^{***}$ | -3.97$^{***}$ | -0.16 (ns) | 1.45 (ns) | **3.39**$^{***}$ | **2.41**$^*$ | 0.71 (ns) |
| | Qwen3-8B | -3.08$^{**}$ | -2.75$^{**}$ | -3.16$^{**}$ | 0.89 (ns) | **4.99**$^{***}$ | **2.56**$^*$ | 1.19 (ns) |
| Clarification | Llama-8B-Instruct | **4.97**$^{***}$ | **3.59**$^{***}$ | **6.45**$^{***}$ | 1.74 (ns) | **5.54**$^{***}$ | **6.96**$^{***}$ | **4.35**$^{***}$ |
| | Mistral-7B | -0.70 (ns) | **2.24**$^*$ | **4.58**$^{***}$ | **-2.75**$^{**}$ | **4.46**$^{***}$ | **6.79**$^{***}$ | 0.54 (ns) |
| | Qwen3-0.6B | **8.53**$^{***}$ | **8.30**$^{***}$ | **5.45**$^{***}$ | **-6.53**$^{***}$ | **10.48**$^{***}$ | **6.47**$^{***}$ | **10.23**$^{***}$ |
| | Qwen3-4B | 1.29 (ns) | -0.36 (ns) | -0.24 (ns) | -1.09 (ns) | **2.44**$^*$ | **3.04**$^{**}$ | **2.04**$^*$ |
| | Qwen3-8B | 1.71 (ns) | -0.65 (ns) | **2.28**$^*$ | **-2.43**$^*$ | **3.86**$^{***}$ | **5.83**$^{***}$ | **3.30**$^{***}$ |

**Experiment 2: GSS predicts when a model asks clarification questions**    Even when a prompt is ambiguous, LLMs do not always ask for clarification, but the field currently lacks an understanding of why models do not seek clarification. As a first step towards such an understanding, it would be useful to be able to predict whether a model would ask a clarification question for a given prompt. Using the different metrics introduced above, we examine when LLMs ask for clarification questions. For each ambiguous prompt, we collected 10 responses from each model and used GPT-4o to annotate whether any of the 10 responses contained at least one clarification question. Then, we examined whether the metric scores are significantly higher when LLMs ask a clarification question — meaning that the scores encode information about a model's clarification behaviors. We find that while most metrics are somewhat informative, $E_{\text{output}}$ and $E_{\text{average}}$ are the only metrics with statistically significant difference between prompts that triggered clarifications and prompts that do not across all models (Table 3).

These results reveal that EigenScore is correlated with not only whether prompts are ambiguous but also whether the models themselves actually output clarification questions in response to these ambiguous questions. Along with EigenScore's high performance on GSSBench, this finding further corroborates that EigenScore, and GSS more broadly, provides interpretable insights into model behaviors.

## 4.2 MEASURING RASONING MODELS' GSS TO ADDRESS REASONING MODEL FAILURES

Building on prior work using UQ metrics to improve the performance of reasoning models (Fu et al., 2025; Kang et al., 2025), we hypothesize that GSS can also predict and improve accuracy on reasoning tasks. We view two failure modes of reasoning models (Sui et al., 2025) under the lens of generation space: when they "overthink" and generate excessive reasoning tokens for simple problems (Liu et al., 2024), their GSS is too large; when they "underthink", generating insufficient reasoning tokens for difficult problems (Su et al., 2025), the models' GSS is too small. To empirically demonstrate the utility of GSS in addressing these issues, we first examine whether our metrics can capture a reasoning model's GSS, in particular the number of possible solution paths to a problem. Then, we show the connection between GSS and reasoning token length, a good proxy for task difficulty (de Varda et al., 2025).

**Experiment 1: GSS measures the number of solution paths**    Following our design for the Random Choice dataset in GSSBench, we construct prompt pairs $(p, p')$ where $p'$ has more possible solution paths than $p$. Specifically, for 1000 logic questions randomly sampled from the Big Reasoning Traces dataset (Allen Institute for AI, 2025), we used GPT-4o to come up with 5 possible solution paths. Then, prompt $p$ is designed to contain only one solution path, constraining the model's choice, while prompt $p'$ contains 5 paths, a wider set of possibilities, allowing the model to choose any one of the 5. The contrast between $p$ and $p'$ yields $|G_t(p')| > |G_t(p)|$. As on GSSBench, we evaluate the pairwise accuracy for each metric $f$. We find that $E_{\text{output}}$ achieves the highest accuracy across all models (and

is significantly higher than any other metric for Qwen3-4B and Qwen3-8B), suggesting that it is a good proxy for reasoning models' GSS. For each metric, all models have comparable performance.

Table 4: **Pairwise accuracy of each metric on the reasoning tasks with specifications of broader versus narrower solution paths.** All error bars are within 0.03. The metric with the highest accuracy for each reasoning model is in **bold**, and the reasoning model with the highest accuracy for each metric is *italicized*.

| Model | Perplexity | Energy | Norm. Entropy | Lex. Sim. | $E_{\text{output}}$ | $E_{\text{average}}$ | Sem. Entropy |
|---|---|---|---|---|---|---|---|
| Qwen3-0.6B (R) | 0.55 | 0.55 | 0.59 | 0.60 | **0.65** | 0.46 | 0.55 |
| Qwen3-4B (R) | *0.61* | 0.62 | 0.63 | 0.60 | ***0.73*** | *0.57* | *0.58* |
| Qwen3-8B (R) | *0.61* | *0.66* | *0.66* | *0.62* | ***0.73*** | 0.55 | 0.56 |

**Experiment 2: GSS is correlated with reasoning token length** Reasoning token length is related to reasoning models' performance (Levy et al., 2024) and can predict the difficulty of a task, aligning with human effort (de Varda et al., 2025). However, we currently do not understand *when* and *why* models generate longer or shorter tokens. Based on human studies (Ericsson & Simon, 1980), we expect tasks with larger generation spaces to require more reasoning effort[1]. We provide empirical evidence of this link by showing that GSS (as well as other uncertainty quantification metrics more broadly) can predict reasoning token length. As in previous work (Olson et al., 2018), we take the length of the reasoning stream to be indicative of task difficulty and reasoning effort required for a task, and we expect the GSS for such tasks to be larger. To test this, we use two datasets of reasoning tasks: 1. a modal and conditional reasoning dataset (Holliday et al., 2024), and 2. an epistemic reasoning dataset (Suzgun et al., 2024).[2] For each prompt, we obtain the reasoning traces from three reasoning models, Qwen3-0.6B (R), Qwen3-4B (R), and Qwen3-8B (R). We calculate the length of these traces by summing the number of reasoning tokens used. We find that there is a moderate to strong positive correlation between the almost all metrics and the number of the reasoning tokens on these deductive tasks (see Figure 3), and the pattern doesn't hold for other non-deductive tasks, where longer traces are not necessarily associated with larger GSS (see Appendix D.2). Additionally, we conducted an analysis of the correlations across different deductive tasks within the modal logic dataset and found that although the overall correlation is positive, the correlations are negative for some controversial conditional and modal reasoning tasks, where longer verbalization does not correspond to a bigger space representation. We present an additional experiment in Appendix D.3) of directly applying GSS to understand reasoning model failures: we measure how GSS captures model failures on CoT versus zero-shot versions of the same problem. More broadly, our findings provide insight into how reasoning model behaviors relate to models' internal task representations.

---

[1]Note that longer traces can also reflect reasoning inefficiency (Sui et al., 2025), and high cognitive load could also lead to the absence of verbalization in humans. Despite these factors, we expect there to be a general correlation between reasoning token length and generation space, given the existing connection between reasoning and the nature of the tasks (Sprague et al., 2024; Liu et al., 2024; Aggarwal et al., 2025).

[2]Holliday et al. (2024) and Suzgun et al. (2024) are recent high-quality datasets that incorporate insights from contemporary semantic theory, modal logic, and epistemic logic, making them apt for evaluating reasoning abilities across tasks of varying difficulty.

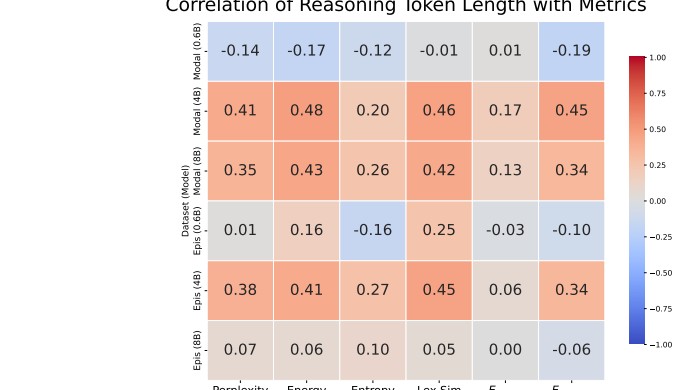

Figure 3: Pearson's $r$ correlation between reasoning token length and various metrics across two datasets (Modal is short for Modal Logic; Epistemic is short for Epistemic Logic), and three Qwen3 model sizes.

### 4.3 EXPANDING GSS: LEAVE-ONE-OUT EIGENSCORE

To address the problem of homogeneity in LLM outputs, we show that steering models to expand their GSS produces high-quality, diverse outputs. Specifically, we explore how EigenScore – the best proxy for GSS thus far – can be used to steer models for this task. Building on DivPO (Lanchantin et al., 2025), which selects the most diverse response from a pool of high-quality responses as the chosen response and the least diverse one in a pool of low-quality responses as the rejected one to perform Direct Preference Optimization (DPO) (Rafailov et al., 2023), we explore applying a similar approach using EigenScore as the diversity criterion.

Since existing forms of EigenScore are for a given *prompt*, we construct a new form of EigenScore as a diversity metric for an individual *response* to measure how much a single generation contributes to the overall spread. Let $S = \{x_1, x_2, \ldots, x_n\}$ denote the set of $n$ sampled responses for a given prompt. We can calculate a single EigenScore across the samples, which we call $E_{\text{global}}$. Now, we define the **Leave-One-Out Eigenscore** (LOOE) metric for response $i$ as:

$$\text{LOOE}_i = E_{\text{global}} - E_i, \text{where } E_i = \text{E}(S \setminus \{x_i\}),$$

i.e., $E_i$ is calculated by removing the response's embeddings from the covariance matrix and recalculating the EigenScore. LOOE is response-centric (provides a score for a particular output rather than a prompt); is semantically aware (operates in meaning space rather than token space); and relies on model internals rather than post-hoc sampling. It is the first diversity metric to have all three of these qualities (see Table A29 for a comparison of existing diversity metrics' properties).

**Experimental Setup** Since expanding the GSS is primarily critical for open-ended questions such as creative generations, we use prompts with the intent label of *Seek Creativity* from Wang et al. (2024b) and creative prompts from PRISM (Kirk et al., 2024) (filtered using GPT-4o) as training and test data (performing a 0.8-0.2 train-test split, resulting in 1532 training data). We compare against the following baselines: different temperature values ($t = 0.5, 1, 2, 3$); a vanilla DPO model not optimized for diversity (where the model is fine-tuned on preference pairs such that the chosen response is the one with the highest reward, scored by a reward model ArmoRM (Wang et al., 2024a)); the original DivPO implementation using negative log likelihood (NLL); and using lexical similarity as the diversity metric [3].

---

[3] Here, the most diverse response is the one with the greatest distance to the mean of the sample embeddings

Table 5: **Comparison of baseline models, the vanilla DPO model, and DivPO with different diversity metrics including LOOE**. Unique 1-grams and entropy are normalized to $[0, 1]$. We set the temperature to 1 for all DPO models. We report results using the best-performing threshold value for each metric (see ablations across threshold value in Table A31).

| Model | $E_{\text{average}}$ ↑ | Lexical Diversity ↑ | Unique 1-grams ↑ | Compression Ratio ↑ | Entropy ↑ | Reward ↑ |
|---|---|---|---|---|---|---|
| Temp 0.5 | -2.488 | 0.151 | 0.185 | 0.240 | 0.871 | 0.114 |
| Temp 1 | -2.431 | 0.184 | 0.222 | 0.290 | 0.871 | 0.114 |
| Temp 2 | -2.322 | 0.254 | 0.312 | 0.372 | 0.890 | 0.108 |
| Temp 3 | -2.165 | 0.349 | 0.392 | 0.423 | 0.914 | 0.084 |
| Vanilla DPO | -2.479 | 0.184 | 0.268 | 0.311 | 0.894 | 0.126 |
| DivPO + NLL ($p$=0.3) | -2.380 | 0.226 | 0.294 | 0.367 | 0.889 | 0.124 |
| DivPO + LOOE ($p$=0.6) | -2.341 | 0.320 | 0.324 | 0.380 | 0.883 | 0.114 |
| DivPO + Lex Sem ($p$=0.6) | -2.416 | 0.286 | 0.316 | 0.364 | 0.884 | 0.119 |

**Results**   DivPO using LOOE achieves similar diversity and reward as using other diversity metrics (Table 5), underscoring EigenScore's utility in capturing GSS. Moreover, it offers more interpretability due to the aforementioned benefits of LOOE: it simultaneously uses information from a model's internal representations of spread (lexical similarity is post-hoc), captures semantics (NLL only captures surface-level diversity), and isolates the contribution of each response to diversity.

Additionally, while Vanilla DPO appears comparable to the baseline in diversity on existing metrics like n-gram count and lexical diversity, $E_{\text{average}}$ is the only metric on which Vanilla DPO is meaningfully lower than the baseline. This suggests that $E_{\text{average}}$ is not only useful for steering but can also be a more informative diagnostic for models' representational diversity. Future work can explore other training paradigms that directly leverage LOOE or EigenScore as signals in online training to make models GSS-aware.

## 5   RELATED WORK

**Uncertainty Quantification and Model Calibration**   Traditionally, confidence calibration in LLMs refer to the alignment between UQ metrics and correctness on questions with ground truth answers, such as factualQA (Huang et al., 2024; Vashurin et al., 2024; Detommaso et al., 2024; Zhao et al., 2024; Shi et al., 2025; Liu et al., 2025). Various approaches, such as semantic entropy (Kuhn et al., 2023; Farquhar et al., 2024; Nikitin et al., 2024), Kernel Language Entropy (Nikitin et al., 2024), and Semantically Diverse Language Generation (SDLG) (Aichberger et al., 2024), have been used to quantify the predictive uncertainty in LLMs to detect hallucination. Other existing work establish a connection between prompt ambiguity and leverage UQ metrics to estimate the aleatoric semantic uncertainty (Aichberger et al., 2024), predict prompt ambiguity in factualQA tasks (Min et al., 2020; Zhang & Choi, 2021), and improve a model's calibration (defined as alignment between UQ metrics and correctness) (Huang et al., 2024; Vashurin et al., 2024; Detommaso et al., 2024; Zhao et al., 2024; Shi et al., 2025; Liu et al., 2025), instructing models to abstain from generating responses (Kamath et al., 2020; Ren et al., 2022; Zablotskaia et al., 2023; Hou et al., 2023) or asking clarification questions if a question is too ambiguous (Cole et al., 2023). Our work focuses on ambiguity in broader use cases rather than only factual QA.

**Diversity Metrics**   Traditional diversity metrics like unique n-gram count cannot distinguish between surface-level variations and functional diversity. Other diversity metrics (e.g. self-BLEU, type-token ratio, compression ratio, linguistic diversity (Guo et al., 2024), and more recently Novelty-Bench (Zhang et al., 2025) and effective semantic diversity (Shypula et al.)) are post-hoc, quantifying variation at the output level without taking into account the model's internal representation. Shypula et al. introduces effective semantic diversity that measures the semantic diversity among high-quality generations for code generation and show that post-trained models actually generate more semantically diverse contents. Zhang et al. (2025) is another attempt to evaluate LLMs for their functional diversity. Steering methods, such as Ismayilzada et al. (2025) and Li et al. (2025), optimize for higher diversity using existing metrics by maximizing diversity measured from output signals. EigenScore (specifically LOOE) as a diversity metric builds upon these previous work to simultaneously offer insight into individual responses; semantic interpretation; and insight into model internals.

## 6 Discussion and Future Work

Like the opaque nature of human thoughts, the GSS of a language model is not readily accessible. Using GSSBench, we provide the first framework to quantify different metrics' ability to represent GSS. We find that EigenScore — a metric that captures the differential entropy in the sentence embedding space (and thus retains rich semantic information) — performs best, highlighting its broader representational power beyond its previously reported hallucination detection capabilities. We encourage future work to use GSSBench to find even better proxies and evaluate more models, especially to investigate the inverse scaling effect (i.e., larger instruction-tuned models are less calibrated to real-world probabilities). GSSBench allows for systematic examination of model's miscalibration of GSS beyond existing diversity metrics, surfacing not only surface-level output homogeneity but a deeper mismatch between real-world distributions and model's internal task representation. We show that various challenges can be tackled under the lens of GSS, and our work lays the foundation for at least three promising future directions: (1) improving an LLM's ability to establish grounding in response to prompt ambiguity (2) since we show a connection between GSS and reasoning model miscalibration, future work can use GSS to address over- and underthinking problems and align a reasoning model's GSS with a task's true GSS (3) developing GSS-aware alignment techniques: having unified factualQA and open-ended generations under the joint problem of GSS miscalibration, an exciting direction of future work is training and aligning models to dynamically adjust their GSS based on different task types, constraining it or expanding it depending on the task.

One key limitation is that GSS is agnostic to the content of the generations. For example, consider a model $m$ that consistently generates the same wrong answer to a factual QA prompt $p$, making its GSS identical to the ground-truth generation space size (both singleton). While we have demonstrated the impressive mileage that we can get out of GSS, we encourage future work to see how GSS can be unified with content-sensitive understandings of model internals. Another limitation is that we observe that $E_{average}$ and $E_{output}$ are not particularly good at the random choice dataset, while semantic entropy has almost perfect accuracy, possibly because $E_{average}$ and $E_{output}$ are more suited for long-form generations.

## 7 Ethical Statement

While we use the "what comes to mind" analogy to motivate why we are interested in exploring the space of possible generations for a language model, we do not wish to anthropomorphize machine cognition, since this notion of "what comes to mind" requires a different empirical investigation than what has been traditionally done in cognitive science for probing human cognition (Ibrahim & Cheng, 2025).

In addition, while we investigate reasoning token length and use it to represent the amount of deliberation required, we acknowledge that reasoning traces are very different from how humans produce thoughts and may not reflect the helpful information that current reasoning literature has taken granted for (Kambhampati et al., 2025).

## 8 Reproducibility Statement

We confirm that our work is reproducible and release our datasets and code. We adapt our implementation of $E_{original}$, $E_{output}$, and $E_{average}$ based on the code used in Chen et al. (2024), and our implementation of semantic entropy is adapted from the repository of Kuhn et al. (2023). We made the following changes: 1. we adjusted the data processing pipeline to adapt to any custom dataset 2. we adjusted the tokenization and inference codes. We open-source all software used in the project and release the datasets used for evaluation.

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

# A  GSSBENCH DETAILS

## A.1  DATASET CONSTRUCTION DETAILS

**Complement**  We generated the base prompts following templates about email, poem, Python program, short story, and persona generation. Each prompt is constructed following an existing template that adds modifiers to the item generation (full details below). Then, the complement version of the prompt is constructed by adding "anything that is not". Tab A2 shows some examples of the prompt pairs.

Table A1: The template used for the Complement dataset. Each base prompt is constructed by choosing a combination of a topic, context, qualifier, and outline

### (a) **An email**

| Field | Example values |
|---|---|
| Topics | job opportunities; an upcoming conference; a new product launch; a team milestone |
| Contexts | at a tech firm; for remote engineers; in the non-profit sector |
| Qualifiers | includes a discussion of my qualifications; asks about remote-work policies |
| Outlines | Greeting, Purpose, Qualifications, Next steps; Subject, Body, Closing |

### (b) **A poem**

| Field | Example values |
|---|---|
| Topics | autumn leaves; lost love; a starry night; the ocean's whispers |
| Contexts | in a small town; during wartime; over the desert |
| Qualifiers | employs vivid imagery; uses iambic pentameter; is limited to 14 lines |
| Outlines | haiku (5-7-5); limerick; free verse |

### (c) **A Python program**

| Field | Example values |
|---|---|
| Topics | sorting a list; scraping a website; converting CSV to JSON; analyzing text sentiment |
| Contexts | using merge sort; handling pagination; with nested objects |
| Qualifiers | includes docstrings; uses type hints; avoids external libraries |
| Outlines | main(), helper functions, guard block; CLI interface |

### (d) **A short story**

| Field | Example values |
|---|---|
| Topics | a time-travel mishap; an unlikely friendship; a dystopian future; a family reunion |
| Contexts | in Victorian London; between a robot and a child; ruled by algorithms |
| Qualifiers | written in first person; contains a twist ending; under 500 words |
| Outlines | Freytag's pyramid; journal entries; letters format |

### (e) **A persona**

| Field | Example values |
|---|---|
| Topics | a tech-savvy college student; a health-conscious parent; a budget traveler; a small business owner |
| Contexts | majoring in computer science; with two toddlers; backpacking in Southeast Asia |
| Qualifiers | includes demographic info; identifies pain points; lists preferred communication channels |
| Outlines | Background, Goals, Challenges; bullet points; short narrative example |

Table A2: Examples of original prompts and their complement versions for the Complement Dataset.

| Original Prompt | Complement Prompt |
|---|---|
| Generate a poem about the moon | Generate anything that is not a poem about the moon |
| Generate a story set in a dystopian future | Generate anything that is not a story set in a dystopian future |
| Generate a Python function to sort a list | Generate anything that is not a Python function to sort a list |
| Generate an email to request a recommendation letter | Generate anything that is not an email to request a recommendation letter |
| Generate a recipe using only 5 ingredients | Generate anything that is not a recipe using only 5 ingredients |
| Generate a haiku about the ocean | Generate anything that is not a haiku about the ocean |
| Generate a motivational quote | Generate anything that is not a motivational quote |
| Generate a summary of the French Revolution | Generate anything that is not a summary of the French Revolution |

**FactualQA Synthetic**  The synthetic dataset for question pairs where one question has one single correct answer and the other has multiple correct answers is constructed using a template with a superlative version of the question and a non-superlative one. To augment the dataset, we populated

variables like country or continent with a randomly selected country or continent name from a pool of candidates. The full prompt template pairs and the country and continent candidates are in Tab A3. We used a total of 60 base prompts, 30 country names, and 6 continent names to populate 1000 unique prompt pairs for evaluation.

Table A3: Templates used to construct the factualQA Synthetic dataset.

(a) Example template pairs. Prompt A has a smaller generation space size than prompt B.

| Prompt A | Prompt B |
|---|---|
| Who was the first president of {country}? | Name a president of {country}. |
| What is the capital of {country}? | Name a city in {country}. |
| What is the largest river in {country}? | Name a river in {country}. |
| What is the tallest mountain in {country}? | Name a mountain in {country}. |
| What is the longest river in {continent}? | Name a river in {continent}. |
| What is the most populated city in {country}? | Name a city in {country}. |
| What is the highest mountain in {continent}? | Name a mountain in {continent}. |
| What is official language of {country}? | Name a language spoken in {country}. |
| What is the currency of {country}? | Name a currency used in {continent}. |
| Who was the 16th president of the United States? | Who was a president of the United States? |

(b) Countries and continents to replace the placeholder.

| Type | List |
|---|---|
| Countries | Argentina, Australia, Bangladesh, Belgium, Brazil, Canada, Chile, China, Colombia, Denmark, Egypt, Ethiopia, Finland, France, Germany, India, Indonesia, Iran, Iraq, Italy, Japan, Kenya, Mexico, Netherlands, Nigeria, Pakistan, Russia, South Africa, South Korea, United Kingdom |
| Continents | Asia, Africa, Europe, North America, South America, Australia |

Table A4: Example categories and their items used to construct synthetic prompts for the random choice experiment.

| Category | Items |
|---|---|
| Animals | cat, dog, sheep, horse, bird, whale, lion, tiger, bear, elephant, giraffe, zebra |
| Colors | red, blue, green, yellow, black, white, orange, purple, pink, gray, brown, cyan |
| Numbers | 1, 2, 3, 4, 5, 6, 7, 8, 9, 10, 11, 12, 13, 14, 15, 16, 17, 18, 19, 20 |
| Fruits | apple, banana, cherry, grape, kiwi, lemon, mango, orange, pear, peach, plum, melon |
| Vehicles | car, truck, bus, motorcycle, bicycle, scooter, van, train, boat, plane, helicopter, submarine |

**Random Choice**   To construct the prompt pairs for the random choice experiment, we used a word bank from four categories: animals, colors, numbers, and vehicles. Each category contains 10 to 20 common words. The prompt pairs are constructed by first randomly choosing a category, then randomly choosing 2 (for prompt A) or 10 (for prompt B) items from the list to append to the sentence "Choose one from the following:". The full list of words are in Tab A4. To verify that each option has a similar probability of being chosen and that the space size is truly bigger for the bigger set, we calculate the logits distribution for each question and find that the logits distribution is uniform for the original prompts (with two options), and is relatively uniform for the complement prompts (although some tokens are favored than others, see Tab A5).

Table A5: Examples of token logits for Original (2 tokens) and Expand (10 tokens).

| Prompt | Label | Token Logits |
|---|---|---|
| Choose one from the following: melon, orange, banana, grape, strawberry, apple, cherry, pear, pineapple, peach | expand | [21.09, 20.66, 18.08, 18.78, 20.66, 17.61, 18.70, 19.44, 20.05, 20.33] |
| Choose one from the following: truck, bus, subway, bicycle, motorcycle, van, car, scooter, tram, train | expand | [21.22, 20.16, 22.67, 20.11, 20.00, 20.64, 17.19, 17.72, 15.20, 20.66] |
| Choose one from the following: orange, melon | original | [25.20, 25.28] |
| Choose one from the following: scooter, helicopter | original | [22.41, 25.83] |

**Subset** The subset dataset is constructed by appending additional information (adding additional requirements) to each base generation task. The base generation tasks are the same as the complement dataset: email, poem, Python Program, short story, or persona generation. For each group, we create five prompts of increased specificity level by appending more and more requirements. We evaluate the pairwise accuracy within each set of 5 prompts (with 10 comparisons). The dataset comprises of 180 sets of prompts and a total of 900 prompts. Tab A6 shows an example of a set of prompts, where there are five levels of specificty and 10 pairs of comparisons: specificity, we have the following relationships: $G_t(A) > G_t(B)$, $G_t(A) > G_t(C)$, $G_t(A) > G_t(D)$, $G_t(A) > G_t(E)$, $G_t(B) > G_t(C)$, $G_t(B) > G_t(D)$, $G_t(B) > G_t(E)$, $G_t(C) > G_t(D)$, $G_t(C) > G_t(E)$, $G_t(D) > G_t(E)$.

Table A6: An example set of prompts from Subset Dataset.

| Prompt ID | Prompt |
|---|---|
| A | Write an email |
| B | Write an email about job opportunities |
| C | Write an email about job opportunities at a tech firm |
| D | Write an email about job opportunities at a tech firm that includes a discussion of my qualifications |
| E | Write an email about job opportunities at a tech firm that includes a discussion of my qualifications and follows the outline: 1) Greeting 2) Purpose 3) Qualifications 4) Next steps |

**Union** The union dataset is constructed by taking the union (connecting generation tasks with the keyword "or"), which increases the theoretical generation space (but model often miscalibrates on such prompts). For each group, we create 4 base prompts (e.g. "come up with an idea for breakfast", "come up with an idea for lunch", "come up with an idea for afternoon snack", and "come up with an idea for dinner"), then we create a total of 15 prompts, including each possible combination of the base prompts, connected through "or". We evaluate whether the scores for the bigger sets (e.g. "come up with an idea for breakfast or lunch or dinner or afternoon snack") are bigger using pairwise comparisons. Within each set, there are 15 prompts and 50 comparisons we can make (there are 105 pairs in total, yielding 50 subset-superset relations), following the logic that the size of a set is strictly smaller than or equal to an element in its superset. We created 60 distinct sets.

Table A7: An example set of prompts from Union Dataset.

| Elements | Prompt |
|---|---|
| A | Come up with an idea for breakfast |
| B | Come up with an idea for lunch |
| C | Come up with an idea for dinner |
| D | Come up with an idea for afternoon snack |
| AB | Come up with an idea for breakfast or lunch |
| AC | Come up with an idea for breakfast or dinner |
| AD | Come up with an idea for breakfast or afternoon snack |
| BC | Come up with an idea for lunch or dinner |
| BD | Come up with an idea for lunch or afternoon snack |
| CD | Come up with an idea for dinner or afternoon snack |
| ABC | Come up with an idea for breakfast or lunch or dinner |
| ABD | Come up with an idea for breakfast or lunch or afternoon snack |
| ACD | Come up with an idea for breakfast or dinner or afternoon snack |
| BCD | Come up with an idea for lunch or dinner or afternoon snack |
| ABCD | Come up with an idea for breakfast or lunch or dinner or afternoon snack |

**Intersection** Each group in the intersection dataset comprises of 4 base prompts, which are overlapping requirements (e.g. "compose an email", "please write a piece that is 200 words long", "please write something that is three paragraphs in length", and "compose a piece using formal language"). Then, we can take the intersections by connecting each base prompt with the keyword "and", which effectively constrains the generation space by adding additional requirements. We created 60 unique sets (each with 15 prompts) and evaluate the pairwise comparison based on whether the score for each subset is smaller than the score of its supersets. Again, each set of 15 prompts yields 50 pairs of comparisons based on subset-superset relationships.

| Model | Task | Accuracy (% $\pm$ 1.96 SE) |
|---|---|---|
| GPT-4o | Complement | $100.00 \pm 0.00$ |
| | FactualQA | $100.00 \pm 0.00$ |
| | Intersection | $77.50 \pm 1.47$ |
| | Random Choice | $100.00 \pm 0.00$ |
| | Subset | $99.39 \pm 0.18$ |
| | Union | $99.87 \pm 0.04$ |

Table A9: The agreement between GPT-4o judge and our ground-truth prompt-pair constructions. The prompt used was: *You are an expert judge of eneration space size (the theoretical space of all possible valid generations for a given prompt). For the two prompts below, determine which one has a bigger generation space size (i.e., more possible valid answers). Prompt A:..., Prompt B:... Only output a single character: A if Prompt A has the bigger generation space; B if Prompt B has the bigger generation space.*

| Prompt A | Prompt B |
|---|---|
| Write something using the past tense and include dialogue. | Please write something in the past tense. |
| Write a blog post that ends with a conclusion. | Please write something that concludes with a final statement. |
| Please write content that includes step-by-step instructions along with code examples. | Please provide step-by-step instructions for writing something. |
| Write a movie review that includes mentions of both the director and the soundtrack. | Compose a piece that references the soundtrack. |

Table A10: Examples of disagreement on the intersection dataset. Prompt B has a bigger generation space under our ground truth construction, while GPT-4o annotated prompt A as having a bigger generation space.

Table A8: An example set of prompts from Intersection Dataset. Each prompt is created by taking the intersection of the base prompts.

| Elements | Prompt |
|---|---|
| A | Compose an email. |
| B | Please write a piece that is 200 words long. |
| C | Please write something that is three paragraphs in length. |
| D | Compose a piece utilizing formal language. |
| AB | Compose an email with a word count of approximately 200 words. |
| AC | Compose an email consisting of three paragraphs. |
| AD | Write an email using formal language. |
| BC | Compose a 200-word piece divided into three paragraphs. |
| BD | Compose a piece of writing that contains 200 words, utilizing formal language throughout. |
| CD | Compose a text consisting of three paragraphs, ensuring the use of formal language throughout. |
| ABC | Compose an email that contains 200 words and is organized into three paragraphs. |
| ABD | Compose a formal email with a word count of approximately 200 words. |
| ACD | Compose an email consisting of three paragraphs, written in formal language. |
| BCD | Please write a 200-word text divided into three paragraphs using formal language. |
| ABCD | Compose a formal email consisting of three paragraphs and approximately 200 words. |

## A.2 VALIDATION

To validate the construction of the prompt pairs, we used GPT-4o to annotate the prompt from each pair that has a bigger generation space size and report the results in Tab A9. We find that there is an almost-perfect agreement for all tasks, except for intersection, and report disagreements in the intersection dataset in A10.

## A.3 RESPONSE QUALITY

To verify the quality of the responses, we used GPT-4o to annotate for response validity and report the results in Tab A13.

## A.4 ROBUSTNESS CHECK

To verify that the model calibration error is sufficiently small that model orderings transfer, we performed a robustness check on the Random Choice dataset. For each prompt, we sampled 10 model responses and treated the number of unique generations across the samples as a proxy for the model's GSS and calculated the number of times when the number is greater for the prompt with fewer options to choose from. We found that the violation rate is very small (0.7% for Llama-8B-Instruct, 4.4% for Mistral, 1.1% for Qwen-0.6B, 7.5% for Qwen-4B, and 5% for Qwen-8B). We exclude these instances and re-calculated the accuracy and find that model orderings indeed transfer (see Tab A12).

Table A11: The full results without excluding any low-quality responses.

| Model | Perplexity ↑ | Energy ↑ | Entropy ↑ | Lex Sim ↓ | $E_{\text{original}}$ ↑ | $E_{\text{output}}$ ↑ | $E_{\text{average}}$ ↑ | Sem En ↑ |
|---|---|---|---|---|---|---|---|---|
| Llama-8B-Instruct | *0.600* | *0.587* | *0.612* | *0.665* | 0.535 | 0.717 | ***0.724*** | 0.546 |
| Mistral-7B | 0.395 | 0.558 | 0.464 | 0.608 | 0.487 | 0.595 | **0.630** | 0.497 |
| Qwen3-0.6B | 0.518 | 0.531 | 0.421 | 0.615 | *0.572* | ***0.747*** | 0.648 | *0.578* |
| Qwen3-4B | 0.511 | 0.532 | 0.515 | 0.555 | 0.491 | **0.604** | 0.590 | 0.512 |
| Qwen3-8B | 0.477 | 0.434 | 0.487 | 0.518 | 0.510 | 0.586 | **0.613** | 0.480 |

## A.5 THE EFFECT OF PROMPT LENGTH

Here we provide clarity on the connection between $G_t(p)$ and the length of a prompt in GSSBench. Specifically, we show that the length of a prompt alone is not predictive of $G_t(p)$. We calculate the correlation between $E_{\text{average}}$ and prompt length in our tasks to clearly illustrate that the higher accuracy of EigenScore is not a result of EigenScores being higher for longer prompts. To address the concern that longer prompts contain more information and are correlated with various uncertainty measurements like entropy (Shannon, 1951), we intentionally construct datasets where longer prompts can correspond to both a greater $G_t(p)$ or a smaller $G_t(p)$. For example, in the Subset dataset, longer prompts correspond to a smaller ground-truth GSS within each set, while for Random Choice, Complement, and Union, the longer prompt in a pair is the one with a bigger $G_t(p)$. In the factualQA prompt pairs, the prompts have similar lengths, so prompt length is not a good predictor for the task of modeling generation space size. In Tab A14, we present the correlation between $E_{\text{average}}$ and prompt length, providing evidence that prompt length is not directly related to $E_{\text{average}}$ .

Table A14: Correlation between $E_{\text{average}}$ and prompt length. We show that there is no consistent correlation between prompt length and $E_{\text{average}}$ for different models.

| Dataset | Llama-8B-Instruct | Mistral-7B | Qwen3-0.6B | Qwen3-4B | Qwen3-8B |
|---|---|---|---|---|---|
| Complement | 0.024 | -0.084 | 0.0066 | 0.015 | -0.023 |
| factualQA | -0.23 | 0.029 | 0.058 | 0.25 | 0.17 |
| Random Choice | -0.018 | 0.080 | 0.56 | 0.36 | 0.081 |
| Subset | -0.47 | -0.47 | -0.29 | -0.15 | -0.34 |
| Union | 0.036 | -0.079 | -0.039 | 0.20 | 0.090 |
| Intersection | -0.13 | -0.24 | -0.060 | 0.060 | 0.066 |

## A.6 FULL RESULTS

We present the full results on each dataset in Tab A16. In addition to the five models, we include results for the reasoning version of Qwen3-0.6B and Qwen3-4B.

## A.7 DISTRIBUTION ANALYSIS

**Comparing Metrics** Below we show the distribution of the two classes for Llama-8B-Instruct on FactualQA (Fig A8) and Random Choice (Fig A2), in addition to Complement (as displayed in the main text). Fig A3 shows the distribution across the five specificity levels on the Subset dataset and the differen levels (the number of elements taken the union or intersection of) in the Union and Intersection datasets.

**Comparing Models** GSSBench enables the comparison across models on the same task using the same metric $D$. Here, we compare the calibration of Qwen3-0.6B, Qwen3-4B, and Qwen3-8B on the

Table A12: Accuracy on the Random Choice dataset excluding cases where models generated more unique words for the original condition.

| Metric | Llama | Qwen-0.6B | Qwen-4B | Mistral-7B | Qwen-8B |
|---|---|---|---|---|---|
| Perplexity | $0.685 \pm 0.04$ | $0.548 \pm 0.04$ | $0.871 \pm 0.03$ | $0.482 \pm 0.04$ | $0.548 \pm 0.04$ |
| Energy | $0.582 \pm 0.04$ | $0.722 \pm 0.04$ | $\mathbf{0.957 \pm 0.03}$ | $\mathbf{0.695 \pm 0.04}$ | $0.242 \pm 0.04$ |
| Entropy | $0.851 \pm 0.04$ | $0.399 \pm 0.04$ | $0.844 \pm 0.04$ | $0.695 \pm 0.04$ | $0.552 \pm 0.04$ |
| Lex Sim | $0.365 \pm 0.04$ | $0.212 \pm 0.04$ | $0.081 \pm 0.03$ | $0.320 \pm 0.04$ | $0.302 \pm 0.04$ |
| $E_{\text{original}}$ | $0.827 \pm 0.04$ | $0.764 \pm 0.04$ | $0.796 \pm 0.04$ | $0.604 \pm 0.04$ | $0.718 \pm 0.04$ |
| $E_{\text{output}}$ | $0.731 \pm 0.04$ | $\mathbf{0.911 \pm 0.03}$ | $0.952 \pm 0.03$ | $0.637 \pm 0.04$ | $\mathbf{0.758 \pm 0.04}$ |
| $E_{\text{average}}$ | $0.492 \pm 0.04$ | $0.886 \pm 0.03$ | $0.839 \pm 0.04$ | $0.588 \pm 0.04$ | $0.726 \pm 0.04$ |
| Semantic E | $\mathbf{0.987 \pm 0.01}$ | $0.889 \pm 0.03$ | $0.806 \pm 0.04$ | $0.619 \pm 0.04$ | $0.702 \pm 0.04$ |

| Model | Dataset | Proportion $\pm$ Error |
|---|---|---|
| Llama-8B | Complement | $0.400 \pm 0.04$ |
| | Intersection | $0.888 \pm 0.01$ |
| | QA | $0.842 \pm 0.03$ |
| | RC | $0.980 \pm 0.01$ |
| | Subset | $0.760 \pm 0.02$ |
| | Union | $0.966 \pm 0.01$ |
| Mistral-7B | Complement | $0.143 \pm 0.03$ |
| | Intersection | $0.369 \pm 0.02$ |
| | QA | $0.423 \pm 0.04$ |
| | RC | $0.236 \pm 0.04$ |
| | Subset | $0.429 \pm 0.02$ |
| | Union | $0.507 \pm 0.02$ |
| Qwen-0B | Complement | $0.310 \pm 0.04$ |
| | Intersection | $0.794 \pm 0.01$ |
| | QA | $0.302 \pm 0.04$ |
| | RC | $0.941 \pm 0.02$ |
| | Subset | $0.716 \pm 0.02$ |
| | Union | $0.741 \pm 0.02$ |
| Qwen-4B | Complement | $0.573 \pm 0.04$ |
| | Intersection | $0.959 \pm 0.01$ |
| | QA | $0.659 \pm 0.04$ |
| | RC | $0.935 \pm 0.02$ |
| | Subset | $0.912 \pm 0.01$ |
| | Union | $0.972 \pm 0.01$ |
| Qwen-8B | Complement | $0.585 \pm 0.04$ |
| | Intersection | $0.962 \pm 0.01$ |
| | QA | $0.779 \pm 0.04$ |
| | RC | $0.993 \pm 0.01$ |
| | Subset | $0.916 \pm 0.01$ |
| | Union | $0.987 \pm 0.01$ |

Table A13: Quality validation using GPT-4o. For each model response, we used GPT-4o to determine whether the model response was valid using the prompt: *You are an expert judge of whether the response is valid for a given prompt. A response is considered valid if it answers the question or fulfills the request made in the prompt appropriately. Prompt: XXX, Response: XXX. Output 1 if the response is valid. Output 0 if the response is not valid.*

six datasets using $E_{\text{average}}$ as the proxy for a model's GSS. Fig A4 shows that while Qwen3-0.6B is generally well calibrated on the three tasks, Qwen3-4B and Qwen3-8B confuse the two classes.

**Comparing Miscalibration on Different Tasks**    Finally, for the same mode, GSSBench enables the comparison of calibration across different tasks. We observe that Llama-8B-Instruct miscalibrates on Random Choice but not Complement (see Fig A5). Fig A4 shows that Qwen3-0.6B can clearly distinguish between the two types of prompts using $E_{\text{average}}$ on Random Choice, but not factualQA.

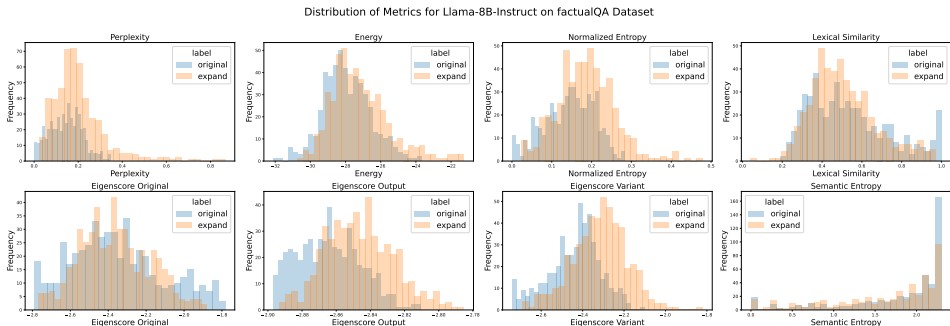

Figure A1: The distribution of metric scores for the two types of prompts for Llama-8B-Instruct on the factualQA Dataset.

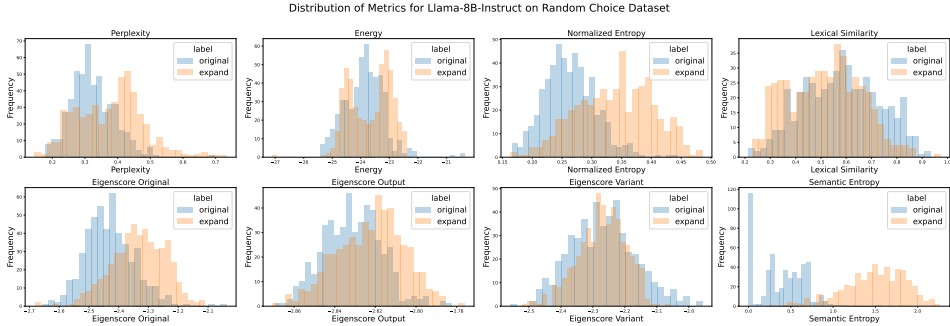

Figure A2: The distribution of metric scores for the two types of prompts for Llama-8B-Instruct on the Random Choice Dataset.

| Category | $E_{\text{average}}$ | Prompt |
|---|---|---|
| Big GSS (Original) | -2.7541 | Generate a short story about an unlikely friendship between rivals that is under 500 words. |
| | -2.7611 | Generate a short story about an unlikely friendship between rivals that contains a twist ending. |
| | -2.7709 | Generate a short story about an unlikely friendship between rivals that is under 500 words and follows Freytag's pyramid: exposition, rising action, climax, falling action, resolution. |
| Small GSS (Complement) | -2.8631 | Generate anything that is *not* a Python program about sorting a list with recursion that uses type hints and is modularized into separate files. |
| | -2.8604 | Generate anything that is *not* a Python program about sorting a list using merge sort that includes docstrings and contains a CLI interface using argparse. |
| | -2.8466 | Generate anything that is *not* an email about job opportunities in the non-profit sector that mentions my attached résumé and follows the outline: 1) Greeting 2) Purpose 3) Qualifications 4) Next steps. |

Table A17: Examples of prompts in the area of overlap on the Complement dataset.

# B ABLATION STUDIES

**Top-K, Sample Size, and Temperature Ablations** We evaluate the role of model parameters such as top-$k$, sample size, and temperature on the Complement Dataset. Consistent with Chen et al. (2024), varying the top-$k$ parameter does not substantially affect performance, while increasing the sample size from 0 to 20 yields steady improvements (Fig A6 and A7). However, we observe that as sample size increases above 20, none of the metrics show significant accuracy improvement, showing that simply increasing the sample size is insufficient in aptly approximating $G_t(p)$. Unlike in hallucination detection, however, EigenScore achieves its best performance on our task at temperature 1.0 rather than 0.5. One possible explanation is that higher sampling randomness produces more diverse embeddings, which may better capture differential entropy when the output space is broader.

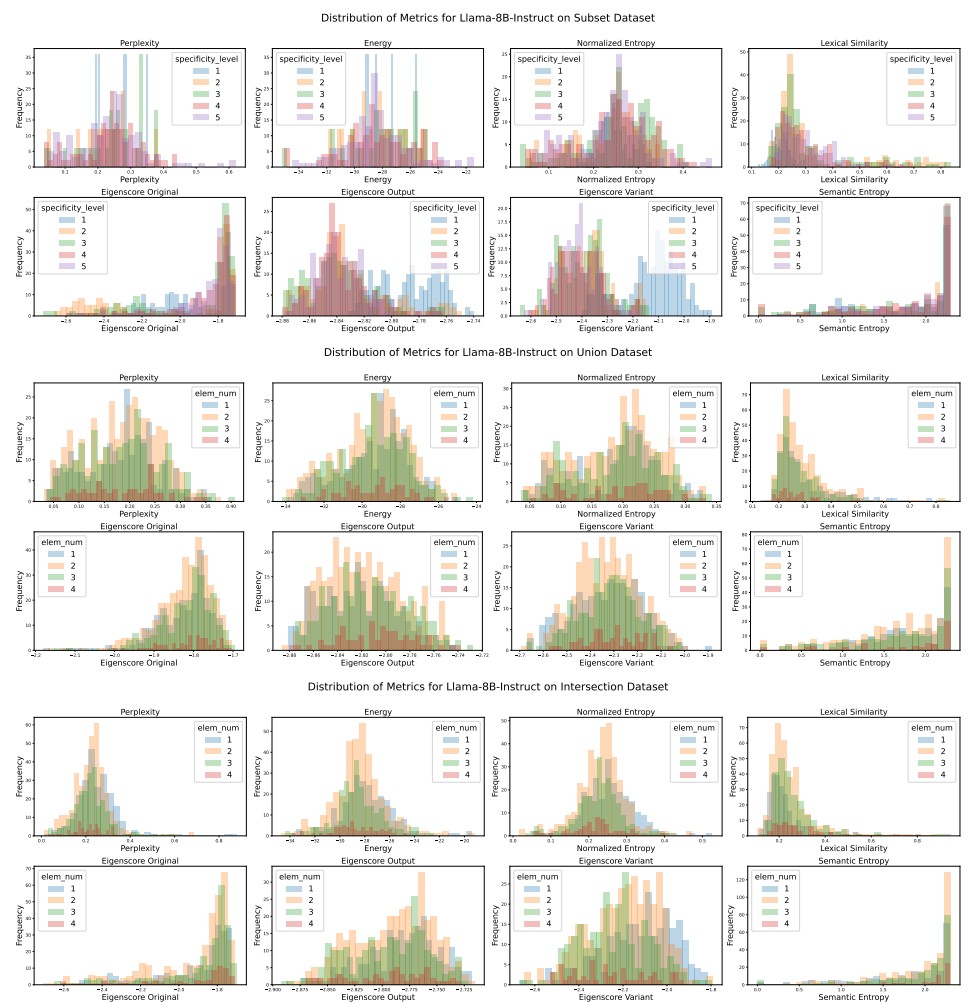

Figure A3: The distribution of metric scores for the two types of prompts for Llama-8B-Instruct. Top: Subset Dataset across different specificity levels (lower means less specific). Middle: Union Dataset across different number of elements being taken in the union (more means greater $G_t(p)$). Bottom: Intersection Dataset across different number of elements being taken in the intersection (more means smaller $G_t(p)$).

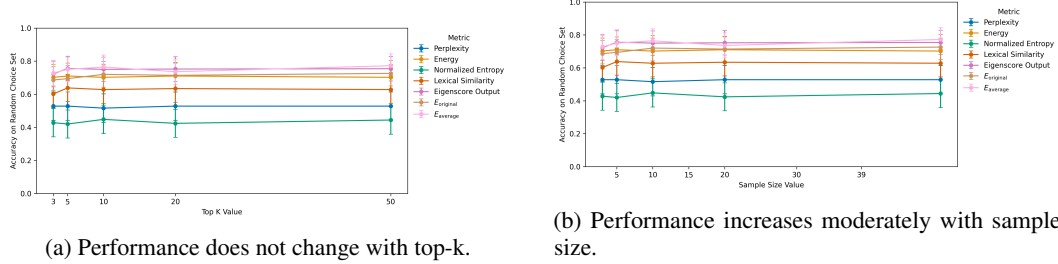

(a) Performance does not change with top-k.

(b) Performance increases moderately with sample size.

Figure A6: Ablation studies on top K and sample size.

$E_{\text{average}}$ **calculation details** There are different ways to implement EigenScore. We perform ablation studies on (1) which layer's embeddings to use and (2) whether to use the last token or average the tokens for the embeddings. We find that individual layers have comparable performance.

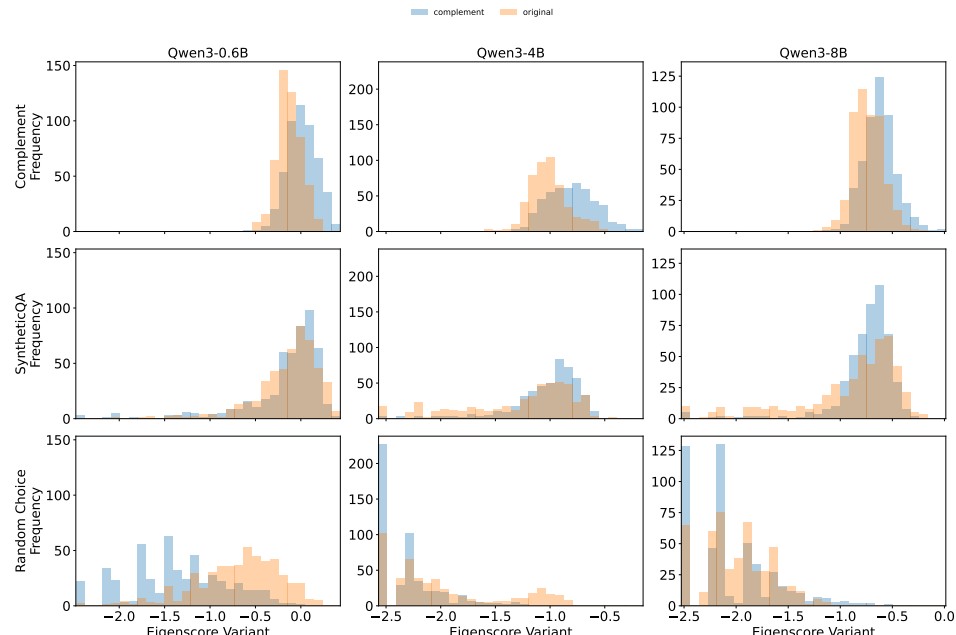

Figure A4: The distribution of $E_{\text{average}}$ across three datasets for Qwen3-0.6B (column 1), Qwen3-4B (column 2), and Qwen3-8B (column 3). Qwen3-4B and Qwen3-8B miscalibrates on the Random Choice dataset, while Qwen3-0.6B doesn't.

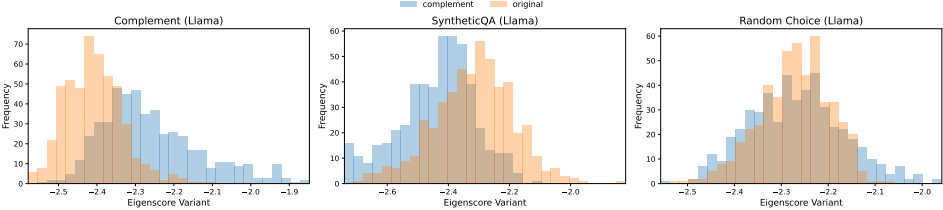

Figure A5: We can use the distributions of $E_{\text{average}}$ on different tasks for the same model to examine its calibration failures on different types of generation tasks. Llama-8B-Instruct can cleanly separate between the Complement classes and the factualQA task but fail the Random Choice task, revealing that its generation space when presented with more options is not aligned with the ground truth generation space.

More critically, taking the mean of the tokens consistently lead to better performance than taking the last token (Figure A9). Thus we use the following variant of EigenScore:

$$E_{\text{average}} \;=\; \frac{1}{|S|\,K} \sum_{\ell \in S} \log \det\!\left( (JZ^{(\ell)})(JZ^{(\ell)})^{\top} + \alpha I_K \right) \tag{7}$$

That is, let $H_{\ell,t}^{(n)} \in \mathbb{R}^d$ denote the hidden state for sequence $n \in \{1, \dots, K\}$, layer $\ell \in \{1, \dots, L\}$, and token $t$; let $T_n$ be the sequence length; define $J = I_K - \frac{1}{K}\mathbf{1}\mathbf{1}^{\top}$ and a small regularizer $\alpha > 0$; and use the layer subset $S = \{20, \dots, L-2\}$. Relative to $E_{\text{original}}$, $E_{\text{average}}$ changes the representation and the aggregation in two ways: (1) for each layer $\ell$ and sequence $n$, replace the single (layer, token) embedding with $\bar{h}_\ell^{(n)} = \frac{1}{T_n-1} \sum_{t=1}^{T_n-1} H_{\ell,t}^{(n)}$ ; (2) for each $\ell$, stack $\bar{h}_\ell^{(n)}$ across sequences to form $Z^{(\ell)}$ to compute the centered covariance, then average the layerwise scores over $S$. Thus, unlike $E_{\text{original}}$'s single-layer, single-token log-det, $E_{\text{average}}$ aggregates over tokens (per layer) and layers.

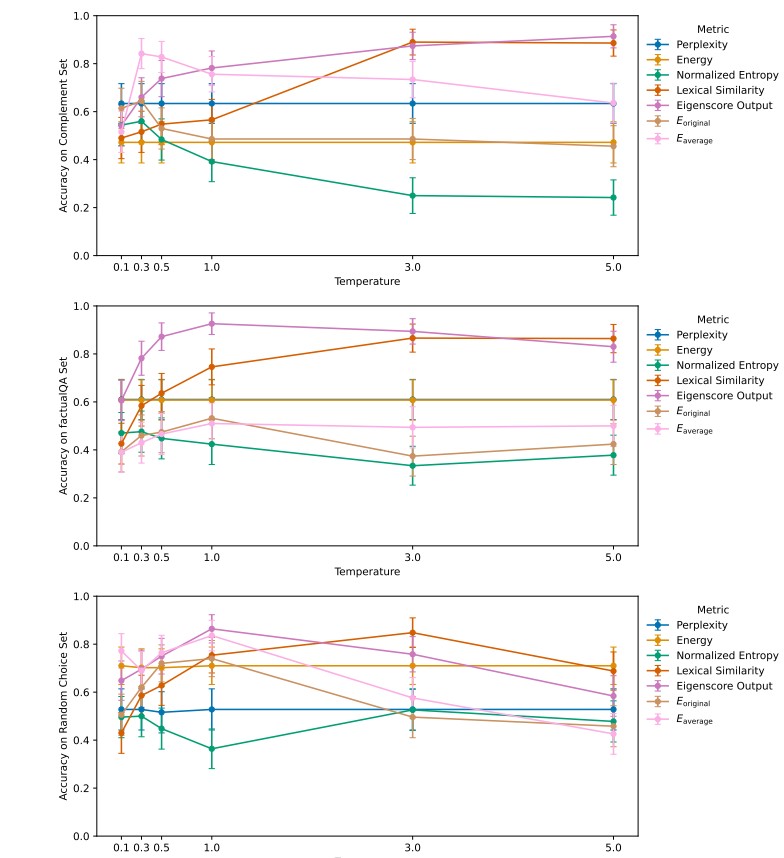

Figure A7: We perform ablation on different temperature values for all metrics on Complement, factualQA, and Random Choice and find that $t = 1$ optimizes accuracy across different metrics.

## C  GROUNDING EXPERIMENT DETAILS

Table A18: Examples of prompts with very low or high $E_{\text{average}}$ scores and their labels from the RIFTS.

| Prompt | Label | $E_{\text{average}}$ |
|---|---|---|
| **Low $E_{\text{average}}$ values** | | |
| Is water wet? (short answer only) | ambiguous | -2.76 |
| How would you go about introducing shading into a 3D game written in C# and Monogame? | none | -2.73 |
| Large tunable lateral shift in prism coupling system containing a superconducting slab is investigated by Yongqiang Kang et al — please edit this statement | advancing | -2.72 |
| Make a markup calculator using HTML, CSS, and JavaScript; results should be displayed in charts | none | -2.71 |
| **High $E_{\text{average}}$ values** | | |
| Please make some comment | addressing | -1.89 |
| Say something out of pocket | ambiguous | -1.90 |
| What's the versions? | ambiguous | -2.04 |
| Do you have photos? | ambiguous | -2.20 |
| Backstory for hazardouslemons | addressing | -2.23 |

Figure A8: The number of unique generations remains stable for the prompts with two options and increases slightly for the prompts with ten options on the Random Choice dataset.

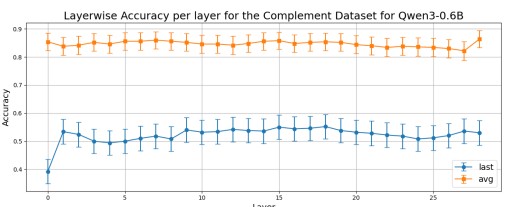

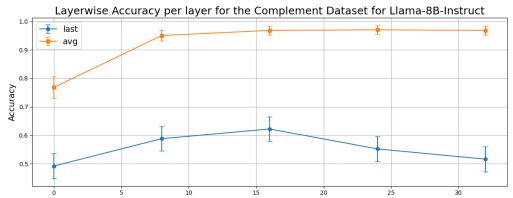

(a) Performance does not change with layer for Qwen-0.6B on the complement dataset. An EigenScore is calculated for each of the 29 layers.

(b) Performance does not change with layer for Llama-8B-Instruct on the complement dataset. An Eigen-Score is calculated for layer 0, 8, 16, 24, and 32.

Figure A9: Ablation studies on the layer to take the embeddings from and the token choice (last token versus averaging all tokens).

## C.1 RIFTS DETAILS

We use **RIFTS**, which contains prompt and grounding-act label pairs [4]. The four possible labels include addressing, ambiguous, advancing, and none [5]. "Addressing" and "ambiguous" are cases where the model or the user has to ask for or provide additional information or clarification, signaling grounding failure, while "advancing" and "none" are prompts that lead to the successful continuation of a conversation. We group the former two as *ambiguous* and the latter as *non-ambiguous* and examine which metrics can separate the two classes to capture a model's representation of ambiguous prompts on everyday generation tasks.

---

[4]The grounding acts are predicted by a forecaster trained on GPT-annotated data of the full human-LLM conversations from WildChat

[5]Advancing acts are conversational acts that signal common ground, which lead to successful next-turn conversations. Disambiguating acts are attempts to present failures like asking for clarification. Addressing acts are repair, reformulation, or restarts that address a lack of common ground in a conversation.

## C.2 AN ADDITIONAL DATASET: FUNNELING VS. FOCUSING

We experiment on a second dataset related to prompt ambiguity. We use a teacher-student interaction dataset with **focusing and funneling** labels (Alic et al., 2022) (focusing encourage students to reflect on their thinking, while funneling insinuates students towards a normative answer), where the focusing prompts or utterances are much more ambiguous than the funneling ones. Since the dataset is not designed for LLMs, we prepend "Imagine you are the student. how would you respond to the following instructor's question?" to the start of the original teacher's utterance to elicit the role-played responses that directly address the original questions. We find that most metrics can distinguish focusing prompts from funneling prompts, showing that it is an easier task.

Table A19: T-test results for the mean of **Funneling vs. Focusing** labels on **Alic et al. (2022)** across models. Values are $t$-statistics. The difference is negative if the mean is greater for the focusing class (since the focusing questions are more open-ended). Stars denote significance levels from $t$-tests (* $p < 0.05$, ** $p < 0.01$, *** $p < 0.001$).

| Model | Perp. | Energy | Norm. Ent. | Lex. Sim. | $E_{\text{original}}$ | $E_{\text{output}}$ | $E_{\text{average}}$ | Sem. Ent. |
|---|---|---|---|---|---|---|---|---|
| Llama-8B-Instruct | **-2.05*** | -1.28 (ns) | **-3.47*** | **6.60*** | -1.45 (ns) | **-4.72*** | **-4.89*** | **-4.72*** |
| Mistral-7B | 0.09 (ns) | -0.01 (ns) | **-2.54*** | **4.29*** | **-5.65*** | **-6.69*** | **-7.72*** | **-2.20*** |
| Qwen-0B | **-3.19*** | **-1.98*** | **-4.40*** | **3.50*** | -0.91 (ns) | **-3.80*** | **-3.74*** | **-3.41*** |
| Qwen3-4B | **-4.96*** | **-6.28*** | -0.83 (ns) | 1.53 (ns) | 2.26* | -0.70 (ns) | -0.51 (ns) | -0.89 (ns) |
| Qwen3-8B | **-2.11*** | **-2.33*** | 0.34 (ns) | **3.68*** | **-3.87*** | **-3.68*** | **-5.00*** | **-4.20*** |

## C.3 CLASSIFICATION TASK ON RIFTS

Table A20: Comparison between the GPT-4o Baseline and the various naive classifiers using the threshold as the cutoff (for Llama-8B-Instruct) on the classification task of distinguishing between ambiguous and non-ambiguous prompts using the dataset from Shaikh et al. (2025).

| Model | Accuracy | Macro-F1 | AUC |
|---|---|---|---|
| GPT Baseline | $0.559 \pm 0.01$ | $0.559 \pm 0.01$ | $0.559 \pm 0.01$ |
| Perplexity (threshold=0.34) | $0.508 \pm 0.02$ | $0.488 \pm 0.02$ | $0.508 \pm 0.02$ |
| Energy (threshold=0.15) | $0.515 \pm 0.02$ | $0.495 \pm 0.02$ | $0.516 \pm 0.02$ |
| Normalized Entropy (threshold=-27.94) | $0.520 \pm 0.02$ | $0.515 \pm 0.02$ | $0.520 \pm 0.02$ |
| Lexical Similarity (threshold=0.35) | $0.533 \pm 0.02$ | $0.515 \pm 0.02$ | $0.503 \pm 0.02$ |
| $E_{\text{original}}$ (threshold = -2.47) | $0.505 \pm 0.02$ | $0.463 \pm 0.02$ | $0.504 \pm 0.02$ |
| $E_{\text{output}}$ (threshold = -2.84) | $0.560 \pm 0.02$ | $0.556 \pm 0.02$ | $0.561 \pm 0.02$ |
| $E_{\text{average}}$ (threshold = -2.45) | $0.565 \pm 0.02$ | $0.557 \pm 0.02$ | $0.565 \pm 0.02$ |

In RIFTS (Shaikh et al., 2025), a forecaster was fine-tuned to predict the grounding act that would occur in a conversation, based on the prompt alone. We define a similar prediction task as a binary classification task to determine whether a prompt would require grounding acts (i.e. the prompts are underspecified) or whether a prompt would advance the conversation without requiring clarification (i.e. prompts are well-structured and specific). We compare the performance between prompting a few-shot classifier using GPT-4o (prompt below) and naive classifiers, where all values above a certain threshold are categorized as ambiguous, and all values below the threshold are categorized as non-ambiguous. We show that even simply thresholding $E_{\text{output}}$ and $E_{\text{average}}$ can lead to comparable performance than the GPT baseline.

**Prompt for GPT Baseline**

Below is the full prompt used for prompting GPT-4o to perform binary classification to categorize ambiguous versus non-ambiguous prompts.

Your goal is to predict whether the next message a user will send would include grounding actions based on their initial instruction to an AI assistant. Namely, you are going to predict whether the initial instruction the user provided provides sufficient grounding for the assistant to respond to the user.

**Message Types**

Here are the two possible categories and definitions.

**name: ADDRESSING OR AMBIGUOUS**

**definition:**

- Grounding actions include addressing and ambiguous acts.

- Addressing acts are made in response to detection of inadequate grounding. They explicitly signal a potential misunderstanding. Here, participants engage with a focus on addressing the failure. This could include rephrasing or repeating their initial query, with little to no change, or explicitly correcting a prior misunderstanding or mistake from the assistant.

- Disambiguating acts represent strategies that participants use to—potentially inefficiently—lower the likelihood of potential misunderstandings, such as clarifications (when a participant seeks to disambiguate an utterance from another participant) or proactively clearing up misunderstandings.

- Examples include follow-up questions like "can you explain this".

- All of the above would be categorized as GROUNDING.

**name: ADVANCING OR NONE**

**definition:**

- Advancing signals understanding, which can include acknowledgements like "I understand".

- A user would continue the conversation, sending a message that does not fit into any of the other categories (addressing or ambiguous).

- None tasks are normally well-specified and factual and do not require any intervention.

- If an initial prompt would not lead to addressing or ambiguous acts, it would be categorized as NONE.

**Examples**

Examples of GROUNDING prompts:

- Blackburn rovers vs West Bromwich albion prediction

- I need to remove a heart

- What causes tailbone pain?

- My friend not want to help me, what to [do] with him?

- What happens when someone quits a job without having another one lined up?

Examples of NONE prompts:

- Write a Main heading about a brand name FFF Digital, which is a digital marketing agency

- Suggest a name for a technical blog consisting of five characters at most, which is compatible with SEO

- 1 week out from my powerlifting meet and I'm not prepared [...] what should I do?

- convert rust String to clap::builder::Str

- Generate a full Harvard references section for the following report: [REPORT]

- Join now, Supplier! or Supplier, Join us! which one is better?

As a reminder, your task is to predict whether a prompt would lead to "GROUNDING" or "NONE" actions in the subsequent conversation (whether they would need to establish grounding in the interaction, after the model response, or whether the prompt requires no additional grounding).

Output 1 if the prediction is GROUNDING, and output 0 if the prediction is NONE.

**Task**

Here is the initial instruction: {row['instruction']}

## C.4 Clarification Experiment Details

**Prompt** We used the following prompt to label whether a prompt elicited a clarification question from an LLM. We found that Llama-8B-Instruct asks the least amount of clarification questions (124 out of 871), while Qwen3-8B asks the most amount of clarification questions (382 out of 871). Tab A21 provides examples of a response with a clarification question and one without.

Determine whether any of the following responses contain a clarification question. A clarification question is when the response asks for more information.

**For example:**

- Input Question: "Generate a poem."

- Clarification Response: "Could you specify what kind of poem I should generate?"

**Instructions:**

- Output **1** if there is at least one clarification question in the 10 responses.

- Output **0** if there are none.

**Question:**
{question_text}
**Responses:**
{response}

Table A21: Examples of prompts with responses that include (left) a clarification question and (right) no clarification.

| Prompt | Qwen3-8B Response (clarification) | Llama Response (no clarification) |
|---|---|---|
| Can AI regulate his bowels | It seems there might be some confusion in your question. Let me clarify: (1) AI cannot regulate human bodily functions; (2) AI can assist in healthcare via analytics, not direct control; (3) AI can control machines/robots, not the human body. **Could you clarify what you meant?** | A rather unique and humorous question! As AI systems don't have a physical body, they don't have a digestive system or bowels. They exist solely in the digital realm and don't require sustenance or waste management. |
| assist me to write a python code, i need the code to help me invest in the egyptian stock. | **Can you clarify what exactly you're trying to do** (e.g., fetch data, backtest, trade, analyze)? Do you have an API key for any financial data source? Are you using a local broker or a global one? | I'd be happy to help you with that! Here's a basic Python script that can help you make informed investment decisions in the Egyptian stock market. Please note that this is a simplified example, and you should always do your own research and consider your own risk tolerance before making any investment decisions. Disclaimer: This script is for educational purposes only and should not be used for actual investment decisions... |
| i can not activate my file in jango | It seems like you're having trouble activating a file in "Jango," but there might be some confusion. **Could you clarify what you're trying to do?** Are you trying to activate a file in a *Django* project? Referring to the (defunct) social media platform "Jango"? Or working with a file that won't activate in some software? | I'm here to help you with your Jango (I assume you meant Django) issue! When you say "I can not activate my file in Jango," I'm going to take a guess that you're trying to run a Django project or app, but you're encountering an issue that's preventing it from activating or running. Here are a few potential solutions to help you troubleshoot: (1) Check your project structure: Make sure your project structure is correct. A typical Django project has the following: manage.py, settings.py, urls.py, wsgi.py, and app directories containing your code... |

## D    REASONING EXPERIMENT DETAILS

### D.1    EXPERIMENT 1

We ranomly sampled 1000 prompts from Big Reasoning Traces (Allen Institute for AI, 2025) and for each prompt, used GPT-4o to generate 5 possible solution paths using the prompt below. Tab A22 shows examples of prompt pairs.

---

Your job is to come up with **5 possible ways** to solve the logic question. You do not need to solve the question; only brainstorm different approaches.

**Example:** If the question is "The sum of 2023 consecutive integers is 2023. What is the sum of the digits of the largest of these integers?", then 5 possible solution paths could be: 1. arithmetic-series formula 2. average 3. pairing symmetry 4. center equals length shortcut 5. shift-by-center method.

**Return your responses in the following format** (separate each path with a space): `1. path1   2. path2   3. path3   4. path4   5. path5`

**Question:** {question_text}
**Response:**

---

Table A22: Examples of paired prompts (PromptA: single method vs. PromptB: multiple methods to choose from).

| PromptA | PromptB |
|---|---|
| Question: The sum of 2023 consecutive integers is 2023. What is the sum of the digits of the largest of these integers? Solve the problem using the following method: arithmetic-series formula | Question: The sum of 2023 consecutive integers is 2023. What is the sum of the digits of the largest of these integers? Solve the problem by using one of the methods below:
1. arithmetic-series formula
2. average
3. pairing symmetry
4. center equals length shortcut
5. shift-by-center method |
| Question: Given $$\tan 2\theta = -2\sqrt{2}, 2\theta \in \left(\frac{\pi}{2}, \pi\right)$$ , find the value of $$\frac{2\cos^2\frac{\theta}{2} - \sin\theta - 1}{\sqrt{2}\sin(\theta + \frac{\pi}{4})}$$ . Solve the problem using the following method: Double angle identity for tangent | Question: Given $$\tan 2\theta = -2\sqrt{2}, 2\theta \in \left(\frac{\pi}{2}, \pi\right)$$ , find the value of $$\frac{2\cos^2\frac{\theta}{2} - \sin\theta - 1}{\sqrt{2}\sin(\theta + \frac{\pi}{4})}$$ . Solve the problem by using one of the methods below:
1. Double angle identity for tangent
2. Trigonometric identities for cosine and sine
3. Half-angle formulas
4. Angle addition formulas
5. Simplification using known values of trigonometric functions |

## D.2 EXPERIMENT 2

Tab A23 shows the dataset used to calculate correlations and the size of each dataset, and Tab A24 shows some examples of prompts and their reasoning token lengths and $E_{\text{original}}$ .

Table A23: The datasets used to examine the correlation with reasoning token lengths.

| Dataset | Source | Size |
|---|---|---|
| Big Reasoning Traces | Allen Institute for AI (2025) | 1000 |
| Modal Logic | Holliday et al. (2024) | 3000 |
| Epistemic Reasoning | Suzgun et al. (2024) | 3000 |

Table A24: Examples of token length and $E_{\text{original}}$ for different prompts from the Modal Logic Dataset. All examples show cases where the prompt with bigger generation space correpond to longer reasoning token length and higher $E_{\text{original}}$ . In the modal logic dataset, uDSmu tasks are significantly more difficult than DS tasks. (The model is Qwen3-8B). The prompt with longer reasoning length and $E_{\text{original}}$ is in **bold** for each pair.

| Task Type | Prompt | Token Len | $E_{\text{original}}$ |
|---|---|---|---|
| DS (Logic) | From "Either the pen is in my bag or it is on my desk" together with "The pen isn't on my desk", can we infer "The pen is in my bag"? | 704 | -1.41 |
| DS (Logic) | From "Either the umbrella is in the car or it tucked away in the closet" together with "The umbrella isn't tucked away in the closet", can we infer "The umbrella is in the car"? | 532 | -1.39 |
| uDSmu (Logic) | Either the cat is napping on the couch or it must be playing in the bedroom. Also, it's not the case that the cat must be playing in the bedroom. Can we infer that the cat is napping on the couch? | **1606** | **-1.21** |
| uDSmu (Logic) | Either the jacket is draped over the chair or it must be hanging in the closet. Also, it's not the case that the jacket must be hanging in the closet. Can we infer that the jacket is draped over the chair? | **1262** | **-1.24** |

**Reasoning Token Length on Everyday Tasks** Wang et al. (2024b) provides prompt and user-intent pairs, where user-intent are labels that each participant reported based on the given taxonomy. The possible labels are: Ask for Advice, FactualQA, Leisure, Seek Creativity, Solve Professional Problem, and Text Assistant. We obtain $E_{\text{original}}$ and the token length for each reasoning models and calculate the average thinking token length and $E_{\text{original}}$ for prompts in each category. Tab A25 shows that categories with longer reasoning token lens, such as `Solve Professional Problem` and `Seek Creativity` also have greater $E_{\text{original}}$ . Similarly, tasks with shorter reasoning token length — including `Ask for Advice` and `FactualQA` — also have lower EigenScores. Tasks from

`Solve Professional Problem` and `Seek Creativity` are more difficult tasks that often require more deliberation. The finding provides evidence for our hypothesis that there is a strong connection between EigenScore, reasoning token length, and the generation space size.

Table A25: Token length and $E_{\text{original}}$ by user intent for data from Wang et al. (2024b) (mean $\pm$ 95% CI). Both EigenScore and reasoning token lengths are calculated for Qwen3-8B. After filtering to only include English prompts, $N = 1000$

| User Intent | Token Len | EigenScore |
|---|---|---|
| Ask for Advice | 298.15 $\pm$ 31.1 | -1.61 $\pm$ 0.02 |
| FactualQA | 295.42 $\pm$ 45.3 | -1.63 $\pm$ 0.02 |
| Leisure | 359.19 $\pm$ 117.6 | -1.59 $\pm$ 0.04 |
| Seek Creativity | 383.09 $\pm$ 132.8 | -1.56 $\pm$ 0.05 |
| Solve Professional Problem | 656.10 $\pm$ 180.9 | -1.50 $\pm$ 0.06 |
| Text Assistant | 328.38 $\pm$ 47.4 | -1.64 $\pm$ 0.05 |

**Reasoning Token Length on Modal and Conditional Reasoning Dataset**   Modal and conditional reasoning tasks differ in difficulty, with some tasks presumably requiring more deliberation than others. With this guiding thought, we categorized all inferences from Holliday et al. (2024) into two classes: Easy and Hard. For instance, we classified simple inference patterns, such as Modus Ponens and Modus Tollens, that students are introduced to in an introductory logic class, as Easy. Inferences that involve operations such as modal distribution over booleans were classified as Hard. Our classification was also guided by the accuracies reported in Holliday et al. (2024); we took it that models have difficulty solving harder tasks and thereby achieve lower accuracies on them. Below we show the average reasoning token length and EigenScore for different tasks based on different difficulty levels, where we group different tasks into easy and hard. Tab A26 shows that the harder reasoning tasks have a longer token length and higher EigenScore.

Table A26: Comparison of Token Length and EigenScore for easy and hard modal and conditional reasoning tasks from the dataset used in Holliday et al. (2024)

| Difficulty Level | Token Len | EigenScore |
|---|---|---|
| Easy | 664.81 $\pm$ 15.39 | -1.19 $\pm$ 0.01 |
| Hard | **1254.93 $\pm$ 59.40** | **-0.96 $\pm$ 0.03** |

Table A27: Token Length and EigenScore per task type.

| Task Difficulty | Task Type | Token Len | EigenScore |
|---|---|---|---|
| Easy | AS | 933.33 $\pm$ 118.50 | -1.10 $\pm$ 0.06 |
| | CONV | 600.05 $\pm$ 37.78 | -1.19 $\pm$ 0.03 |
| | CT | 795.42 $\pm$ 78.99 | -1.19 $\pm$ 0.03 |
| | DA | 621.25 $\pm$ 29.49 | -1.21 $\pm$ 0.03 |
| | DS | 549.66 $\pm$ 20.61 | -1.16 $\pm$ 0.03 |
| | INV | 704.00 $\pm$ 40.22 | -1.24 $\pm$ 0.03 |
| | MP | 441.77 $\pm$ 13.86 | -1.09 $\pm$ 0.03 |
| | MT | 521.69 $\pm$ 21.72 | -1.17 $\pm$ 0.03 |
| | MiN | 728.98 $\pm$ 27.71 | -1.22 $\pm$ 0.02 |
| | NMu | 689.34 $\pm$ 41.07 | -1.24 $\pm$ 0.03 |
| Hard | CMP | **2643.60 $\pm$ 488.00** | **-0.40 $\pm$ 0.05** |
| | DSmi | 1676.39 $\pm$ 108.32 | -0.71 $\pm$ 0.05 |
| | DSmu | 709.02 $\pm$ 44.13 | -1.25 $\pm$ 0.02 |
| | MTmi | 1869.09 $\pm$ 159.29 | -0.50 $\pm$ 0.04 |
| | MTmu | 720.24 $\pm$ 56.45 | -1.24 $\pm$ 0.02 |
| | MuAg | 891.98 $\pm$ 121.42 | -1.25 $\pm$ 0.05 |
| | MuDistOr | 1170.68 $\pm$ 153.26 | -1.12 $\pm$ 0.07 |
| | NSFC | 1018.05 $\pm$ 145.24 | -1.21 $\pm$ 0.07 |
| | WSFC | 934.25 $\pm$ 190.73 | -1.25 $\pm$ 0.05 |

**A negative correlation exists between prompt length and EigenScore on other tasks**   We note that the positive correlation between $E_{\text{original}}$ and reasoning token length is not a result of how $E_{\text{original}}$ is computed. We calculate the correlation between the reasoning token lengths of Qwen3-0.6B, Qwen3-4B, and Qwen3-8B and their $E_{\text{original}}$ and find that $r$ is 0.46, -0.39, and -0.25 for them respectively on the Random Choice dataset, showing that the positive correlation we find in the main

text on the deductive tasks does not hold true for all tasks, showing that the correlation is not because of a general positive correlation between $E_{\text{original}}$ and reasoning token length.

## D.3  ZEROSHOT VS. CoT REPRESENTATIONS

Here, we explore if special instructions in the prompt can affect model representations of a task. For example, for an easy task that requires a straightforward answer, if the model is asked to think step-by-step, does the instruction change its representation of the otherwise easy task, and can we probe this representational shift using the metric candidates for a model's GSS? We experiment with three datasets: a dataset of implicit statistical reasoning tasks (AGL) where overthinking is known to degrade performance in humans and LLMs (Liu et al., 2024); a modal logic dataset (Holliday et al., 2024); and the epistemic reasoning dataset (Suzgun et al., 2024) and experiment with the three Qwen3 models. For each problem, we give the model a zero-shot version (that instructs it to not think too hard) and a chain-of-thought version. We seek examine whether different metrics $D$ can capture the perturbation that the prompt-type brings to the model's implicit representation of how much deliberation a task requires. With this investigation, we seek to explain a curious result in the reasoning space (Liu et al., 2024), where thinking step-by-step deteriorates performance on an easy task (AGL). We hypothesize that the CoT instruction perturbs the model representation of the AGL tasks, which deteriorates performance. Crucially, we think that the CoT instruction should not bring about such deterioration effects on harder tasks like modal logic inferences, which in their representations as hard tasks are represented faithfully. We find that on AGL (where deliberation leads to worse performance), UQ scores are higher for the zero-shot prompts, while on modal logic and epistemic logic prompts, the opposite is true. Further experiments are required to verify the use of UQ metrics to explain reasoning models' task representations under different instructions.

| Dataset (Model) | Perplexity | Energy | Entropy | Lex Sim | $E_{\text{original}}$ | $E_{\text{output}}$ | $E_{\text{average}}$ |
|---|---|---|---|---|---|---|---|
| **AGL Dataset** | | | | | | | |
| Qwen3-0.6B | 0.35 (ns) | -0.37 (ns) | -77.51*** | -1.15 (ns) | -38.36*** | 5.24*** | -19.88*** |
| Qwen3-4B | -13.34*** | -15.87*** | -2.98* | 0.04 (ns) | 8.35*** | 13.80*** | 8.70*** |
| Qwen3-8B | -37.90*** | -85.32*** | -30.87*** | 44.41*** | -51.72*** | -2.92** | -8.05*** |
| **Modal Logic** | | | | | | | |
| Qwen3-0.6B | 48.93*** | 86.61*** | 113.30*** | -70.23*** | 113.89*** | 45.15*** | 667.10*** |
| Qwen3-4B | -32.79*** | -57.59*** | 22.10*** | -5.82*** | -7.36*** | 30.45*** | 13.15*** |
| Qwen3-8B | -99.23*** | -95.89*** | 0.70*** | 68.08*** | -74.21*** | -18.64*** | -78.35*** |
| **Epistemic Logic** | | | | | | | |
| Qwen3-0.6B | 2.35* | 4.75*** | 127.97*** | -52.58*** | 4.27*** | 28.15*** | 25.51*** |
| Qwen3-4B | -20.08*** | -26.47*** | 18.04*** | -15.12*** | -2.46* | 29.45*** | 21.82*** |
| Qwen3-8B | -164.50*** | -191.35*** | -93.29*** | 176.97*** | -159.70*** | -78.43*** | -179.84*** |

Table A28: Comparison of metrics across datasets and for zeroshot vs. cot versions. Stars indicate significance levels (* $p < 0.05$, ** $p < 0.01$, *** $p < 0.001$). For AGL, negative significant values are shown in green, positive significant in red. For Modal Logic and Epistemic Logic, the convention is flipped: positive significant values are shown in green, negative significant in red.

# E  LOOE DETAILS

Tab A29 compares LOO EigenScore with existing diversity metrics; Tab A30 shows an example of the different LOO score assigned to each response and the normalized rewards for each response for the same prompt.

| Metric | Model Internals | Semantically Aware | Response-Centric |
|---|---|---|---|
| Unique N-grams | ✗ | ✗ | ✓ |
| Lexical Similarity | ✗ | ✓ | ✓ |
| Compression Ratio | ✗ | ✗ | ✗ |
| Negative Log Likelihood | ✓ | ✗ | ✓ |
| MC dropout (Folgoc et al., 2021) | ✓ | ✗ | ✓ |
| Semantic Entropy (Kuhn et al., 2023; Farquhar et al., 2024) | ✓ | ✓ | ✗ |
| Effective Semantic Diversity (Shypula et al.) | ✗ | ✓ | ✗ |
| NoveltyBench (Zhang et al., 2025) | ✗ | ✓ | ✓ |
| LOO EigenScore | ✓ | ✓ | ✓ |

Table A29: Comparison of diversity metrics by whether they use model internals (logits, hidden states), are semantically aware (operating in meaning space rather than surface or token space), and whether they are prompt or response centric (or could be adapted to be response-centric).

Table A30: An example of the prompt and 10 samples, with their corresponding Leave-One-Out EigenScore and the normalized rewards. The rewards are normalized where higher LOOE values correspond to higher normalized rewards.

| Prompt | Response | LOO raw | Normalized reward |
|---|---|---|---|
| Hello, how can I help my sister to go vegan - she really want to do so and she has always been very active even as an vegan activist. Ou rmum has a huge issue with it and never supported my decision not to eat meat and now my sister´s. | What a wonderful sibling you are! It's fantastic that your sister is passionate about veganism and wants to make the switch.... 1. **Understand your mom's concerns**... 2. **Educate yourself and your mom**... 3. **Gradual transition**... 4. **Involve your mom in the process**... | -0.026 | 0.23 |
| | It's great that your sister is interested in going vegan and being an activist for animal rights. It's understandable that you're concerned about your mom's potential resistance. Here are some tips to help your sister make the transition to veganism and potentially win over your mom: 1. **Educate yourself and your sister**... 2. **Address your mom's concerns**... 3. **Involve your mom in the process**... 4. **Focus on the positive aspects** | -0.026 | 0.20 |
| | That's great that your sister wants to adopt a vegan lifestyle! It's understandable that your mom might have concerns, but with patience, understanding, and some helpful advice, she might become more open to the idea. Here are some suggestions to help your sister and your mom: **For your sister:** 1. **Educate yourself**:... 2. **Gradual transition**:... 3. **Find vegan-friendly options**: ... 4. **Support network**:... **For your mom:** 1. **Open and honest conversation**:..." | -0.016 | 1 |
| | It's great that your sister is passionate about going vegan and making a positive impact on the environment and animal welfare. It's also understandable that you're concerned about how your mom might react.... 1. **Educate yourself and your sister**:... 2. **Involve your mom in the process**:... 3. **Address your mom's concerns**:... 4. **Lead by example** | -0.024 | 0.37 |
| | What a great sister you are! Helping your sister transition to a vegan lifestyle can be a wonderful and rewarding experience. It's also a great opportunity to bond and share a common goal. I'm happy to offer some tips and suggestions to support your sister and navigate any potential challenges, especially with your mom's resistance.... **Understand your mom's concerns:**... 1. Lack of protein or other essential nutrients 2. Higher costs of vegan alternatives 3. Limited social and cultural opportunities 4. Perceived limitations in meal options **Education and preparation:**... 1. **Provide resources:**... 2. **Encourage gradual transition:** ... 3. **Explore vegan alternatives:** ... 4. **Cook together** | -0.020 | 0.73 |
| | It's great to hear that your sister is passionate about veganism and wants to make a positive change in her lifestyle!... 1. **Educate your mom**: ... 2. **Emphasize personal choice**: ... 3. **Focus on the benefits for your sister**: ... 4. **Offer support and resources**:... | -0.016 | 0.99 |
| | What a wonderful sister you are! Helping a loved one transition to a vegan lifestyle can be challenging, especially when there are opposing views within the family... 1. **Educate yourself**:... 2. **Understand your mom's concerns**:... 3. **Offer support and resources**: ... 4. **Respect your mom's boundaries**: | -0.025 | 0.32 |
| | I'm happy to help you support your sister in her transition to a vegan lifestyle. 1. **Open communication**: ... 2. **Education**: ... 3. **Respect her concerns**: ... | -0.018 | 0.84 |
| | I'm glad to hear that your sister is interested in going vegan! It's a great decision for her health, the environment, and animal welfare... 1. **Understand your mum's concerns**:... 2. **Educate yourself and your mum**: ... 3. **Start with small steps**: ... | -0.029 | 0 |
| | I'm happy to help!... 1. **Educate yourself**:... 2. **Understand her concerns**: ... 3. **Share your experience**: ... 4. **Involve your sister**: ... | -0.021 | 0.61 |

**Detailed Results**

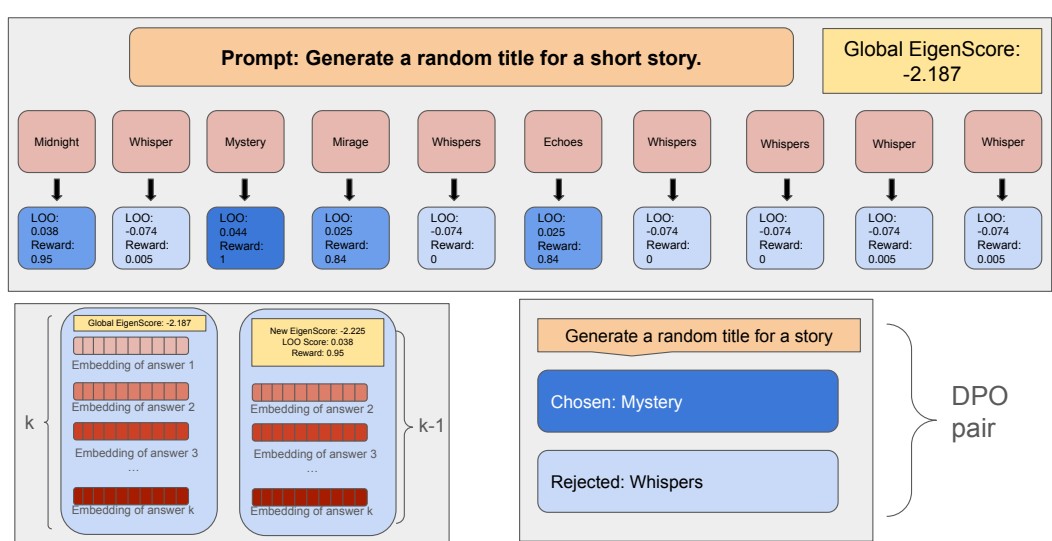

Figure A10: Illustration of the DivPO pipeline (using LOOE as the diversity metric). For each prompt, we first sample 10 generations and calculate the global EigenScore. For each generation, we remove its embeddings from the covariance matrix and re-calculate the EigenScore, and the difference is the LOO EigenScore. We repeat the same process for each response and normalize LOO EigenScore into rewards to construct prefeernce pairs, where the chosen response is the one with the highest LOO EigenScore.

While Lanchantin et al. (2025) trained and evaluated on everyday tasks, we here focus on open-ended tasks where more diverse generations is clearly more desirable. To construct this dataset, we used all 72 prompts with the Seek Creativity label from Wang et al. (2024b) and 1753 open-ended questions in PRISM (Kirk et al., 2024) as the training data to perform DivPO.

We used the following prompt to filter for open-ended tasks from PRISM (Kirk et al., 2024):

Your goal is to categorize whether a prompt is **expand** or **constrain**.
An **expand** prompt is one where it is ideal to have a big generation space, meaning that diverse generations are desired. These include creative tasks, open-ended questions, idea generation, etc., where it is beneficial to have a wide range of possible responses.
A **constrain** prompt, on the other hand, is one where the generation space should be limited, meaning that specific, focused responses are desired. These include tasks that require precise answers, factual information, or specific instructions.
**Definition of expand prompts:**

- Prompts where it is desirable to have diverse generations, like generating random items or creative tasks.

**Definition of constrain prompts:**

- Prompts where it is ideal to have a focused generation space, like generating specific items or factual information. In these cases, we want the responses to be consistent.

- Prompts where the goal is to get a specific answer or information, such as factual questions or requests for specific data or task completion (such as code generation), where we don't care much about diversity of the output.

**Examples of expand prompts:**

- Generate a random number.

- Generate a persona.

- Generate a Python script.

- What hobbies could I do in my spare time?

- My academic advisor is turning 60, and I want to write a song for her birthday. Please help me write some lyrics.

- Write me a unicorn poem.

- Give me a funny pub quiz team name.

- Help me brainstorm possible names for a podcast about musicals in Broadway, movies, TV-shows, and other media.

- Write me a very short screenplay in the style of *Trailer Park Boys*. My name is Steve and I work with Leighton in a lab; we need to work but we bunk off to get drunk.

**Examples of constrain prompts:**

- What city is the hottest in the world?

- When is Singapore independent?

- Can you give me a full list of countries in Eastern Europe?

- Who is Callisto?

- What country has the most oil?

- If electricity usage of 797 gives a refund of 64.41 and usage of 208 gives refund of 1413.67, how much of a refund will there be with usage of 330?

- What is the variance of a variable which has population values of 2, 4, and 6?

As a reminder, your task is to categorize whether a prompt is **expand** or **constrain**.
Output **1** if the prompt is expand, and output **0** if the prompt is constrain.
**Task**
Here is the initial instruction:
{row['user_prompt']}
**Response:**

Table A31: Comparison of baseline models with NLL, LOO, and Lex Sem methods across different threshold values $p$. The bigger the threshold is, the more data included in the pool of candidates (less strict about quality control).

| Model | $E_{\text{average}}$ ↑ | Lex. Div. ↑ | Unique-1g (norm.) ↑ | Comp. Ratio ↑ | Entropy (norm.) ↑ | Reward ↑ |
|---|---|---|---|---|---|---|
| Baseline DPO | -2.480 | 0.184 | 0.268 | 0.311 | 0.894 | 0.126 |
| Temp1 (baseline) | -2.431 | 0.184 | 0.222 | 0.290 | 0.871 | 0.114 |
| NLL ($p = 0.1$) | -2.451 | 0.162 | 0.261 | 0.308 | 0.893 | 0.122 |
| NLL ($p$=0.2) | -2.364 | 0.249 | 0.385 | 0.403 | 0.923 | 0.116 |
| NLL ($p$=0.3) | -2.379 | 0.226 | 0.294 | 0.367 | 0.889 | 0.124 |
| NLL ($p$=0.4) | -2.289 | 0.262 | 0.323 | 0.380 | 0.895 | 0.112 |
| NLL ($p$=0.5) | -2.230 | 0.342 | 0.350 | 0.405 | 0.897 | 0.093 |
| NLL ($p$=0.6) | -2.273 | 0.432 | 0.434 | 0.439 | 0.921 | 0.097 |
| LOO ($p$=0.1) | -2.490 | 0.160 | 0.250 | 0.300 | 0.890 | 0.125 |
| LOO ($p$=0.2) | -2.440 | 0.230 | 0.300 | 0.340 | 0.890 | 0.116 |
| LOO ($p$=0.3) | -2.350 | 0.500 | 0.450 | 0.440 | 0.920 | 0.082 |
| LOO ($p$=0.4) | -2.350 | 0.330 | 0.350 | 0.380 | 0.900 | 0.109 |
| LOO ($p$=0.5) | -2.220 | 0.383 | 0.340 | 0.391 | 0.879 | 0.100 |
| LOO ($p$=0.6) | -2.341 | 0.320 | 0.324 | 0.380 | 0.883 | 0.114 |
| Lex ($p$=0.1) | -2.457 | 0.177 | 0.270 | 0.312 | 0.894 | 0.116 |
| Lex ($p$=0.2) | -2.266 | 0.500 | 0.426 | 0.463 | 0.926 | 0.076 |
| Lex ($p$=0.3) | -2.306 | 0.447 | 0.394 | 0.449 | 0.906 | 0.071 |
| Lex ($p$=0.4) | -2.363 | 0.347 | 0.368 | 0.396 | 0.902 | 0.111 |
| Lex ($p$=0.5) | -2.363 | 0.331 | 0.357 | 0.381 | 0.893 | 0.105 |
| Lex ($p$=0.6) | -2.416 | 0.286 | 0.316 | 0.364 | 0.884 | 0.119 |

Table A15: Accuracy breakdown for each dataset and for each model (without excluding low-quality responses).

### (a) Complement

| Metric | Llama | Qwen-0.6B | Qwen-0.6B (R) | Qwen-4B | Qwen-4B (R) | Mistral-7B | Qwen-8B |
|---|---|---|---|---|---|---|---|
| Perplexity | $0.674 \pm 0.04$ | $0.594 \pm 0.04$ | $0.632 \pm 0.04$ | $0.530 \pm 0.04$ | $0.858 \pm 0.03$ | $0.412 \pm 0.04$ | $0.576 \pm 0.04$ |
| Energy | $0.670 \pm 0.04$ | $0.516 \pm 0.04$ | $0.624 \pm 0.04$ | $0.530 \pm 0.04$ | $0.898 \pm 0.03$ | $0.540 \pm 0.04$ | $0.456 \pm 0.04$ |
| Entropy | $0.772 \pm 0.04$ | $0.354 \pm 0.04$ | $0.352 \pm 0.04$ | $0.690 \pm 0.04$ | $0.778 \pm 0.04$ | $0.314 \pm 0.04$ | $0.532 \pm 0.04$ |
| Lex Sim | $0.880 \pm 0.03$ | $0.668 \pm 0.04$ | $0.716 \pm 0.04$ | $0.736 \pm 0.04$ | $0.704 \pm 0.04$ | $0.560 \pm 0.04$ | $0.712 \pm 0.04$ |
| $E_{original}$ | $0.566 \pm 0.04$ | $0.596 \pm 0.04$ | $0.452 \pm 0.04$ | $0.574 \pm 0.04$ | $0.434 \pm 0.04$ | $0.550 \pm 0.04$ | $0.500 \pm 0.04$ |
| $E_{output}$ | $\mathbf{0.954 \pm 0.02}$ | $\mathbf{0.908 \pm 0.03}$ | $\mathbf{0.958 \pm 0.02}$ | $0.860 \pm 0.04$ | $\mathbf{0.930 \pm 0.02}$ | $0.758 \pm 0.04$ | $0.790 \pm 0.04$ |
| $E_{average}$ | $0.940 \pm 0.02$ | $0.810 \pm 0.03$ | $0.754 \pm 0.04$ | $\mathbf{0.880 \pm 0.03}$ | $0.876 \pm 0.03$ | $\mathbf{0.762 \pm 0.04}$ | $\mathbf{0.806 \pm 0.03}$ |
| Semantic E | $0.492 \pm 0.04$ | $0.692 \pm 0.04$ | $0.336 \pm 0.04$ | $0.562 \pm 0.04$ | $0.200 \pm 0.035$ | $0.5100 \pm 0.04$ | $0.482 \pm 0.04$ |

### (b) SyntheticQA

| Metric | Llama | Qwen-0.6B | Qwen-0.6B (R) | Qwen-4B | Qwen-4B (R) | Mistral-7B | Qwen-8B |
|---|---|---|---|---|---|---|---|
| Perplexity | $0.660 \pm 0.04$ | $0.610 \pm 0.04$ | $\mathbf{0.610 \pm 0.04}$ | $0.318 \pm 0.04$ | $0.428 \pm 0.04$ | $0.086 \pm 0.02$ | $0.334 \pm 0.04$ |
| Energy | $0.656 \pm 0.04$ | $0.608 \pm 0.04$ | $0.486 \pm 0.04$ | $\mathbf{0.410 \pm 0.04}$ | $0.334 \pm 0.04$ | $0.484 \pm 0.04$ | $0.380 \pm 0.04$ |
| Entropy | $0.670 \pm 0.04$ | $0.434 \pm 0.04$ | $0.532 \pm 0.04$ | $0.290 \pm 0.04$ | $0.440 \pm 0.04$ | $0.362 \pm 0.04$ | $0.438 \pm 0.04$ |
| Lex Sim | $0.506 \pm 0.04$ | $0.738 \pm 0.04$ | $0.572 \pm 0.04$ | $0.290 \pm 0.04$ | $0.418 \pm 0.04$ | $\mathbf{0.542 \pm 0.04}$ | $0.274 \pm 0.04$ |
| $E_{original}$ | $0.472 \pm 0.04$ | $0.506 \pm 0.04$ | $0.518 \pm 0.04$ | $0.256 \pm 0.04$ | $0.508 \pm 0.04$ | $0.356 \pm 0.04$ | $0.412 \pm 0.04$ |
| $E_{output}$ | $0.718 \pm 0.04$ | $\mathbf{0.922 \pm 0.02}$ | $0.510 \pm 0.04$ | $0.358 \pm 0.04$ | $\mathbf{0.796 \pm 0.04}$ | $0.280 \pm 0.04$ | $0.388 \pm 0.04$ |
| $E_{average}$ | $\mathbf{0.782 \pm 0.04}$ | $0.502 \pm 0.04$ | $0.556 \pm 0.04$ | $0.284 \pm 0.04$ | $0.606 \pm 0.04$ | $0.468 \pm 0.04$ | $\mathbf{0.438 \pm 0.04}$ |
| Semantic E | $0.370 \pm 0.04$ | $0.320 \pm 0.04$ | $0.500 \pm 0.04$ | $0.352 \pm 0.04$ | $0.392 \pm 0.04$ | $0.474 \pm 0.04$ | $0.372 \pm 0.04$ |

### (c) Random Choice

| Metric | Llama | Qwen-0.6B | Qwen-0.6B (R) | Qwen-4B | Qwen-4B (R) | Mistral-7B | Qwen-8B |
|---|---|---|---|---|---|---|---|
| Perplexity | $0.678 \pm 0.04$ | $0.516 \pm 0.04$ | $\mathbf{0.546 \pm 0.04}$ | $0.696 \pm 0.04$ | $0.654 \pm 0.04$ | $0.464 \pm 0.04$ | $0.458 \pm 0.04$ |
| Energy | $0.594 \pm 0.04$ | $0.702 \pm 0.04$ | $0.452 \pm 0.04$ | $\mathbf{0.762 \pm 0.04}$ | $\mathbf{0.712 \pm 0.04}$ | $\mathbf{0.658 \pm 0.04}$ | $0.312 \pm 0.04$ |
| Entropy | $0.642 \pm 0.04$ | $0.378 \pm 0.04$ | $0.420 \pm 0.04$ | $0.690 \pm 0.04$ | $0.318 \pm 0.04$ | $0.628 \pm 0.04$ | $0.470 \pm 0.04$ |
| Lex Sim | $0.666 \pm 0.04$ | $0.738 \pm 0.04$ | $0.224 \pm 0.04$ | $0.680 \pm 0.04$ | $0.106 \pm 0.03$ | $0.622 \pm 0.04$ | $0.470 \pm 0.04$ |
| $E_{original}$ | $0.680 \pm 0.04$ | $0.726 \pm 0.04$ | $0.510 \pm 0.04$ | $0.618 \pm 0.04$ | $0.656 \pm 0.04$ | $0.562 \pm 0.04$ | $0.542 \pm 0.04$ |
| $E_{output}$ | $0.680 \pm 0.04$ | $\mathbf{0.856 \pm 0.03}$ | $0.236 \pm 0.04$ | $0.704 \pm 0.04$ | $0.550 \pm 0.04$ | $0.600 \pm 0.04$ | $0.562 \pm 0.04$ |
| $E_{average}$ | $0.628 \pm 0.04$ | $0.838 \pm 0.03$ | $0.234 \pm 0.04$ | $0.650 \pm 0.04$ | $0.378 \pm 0.04$ | $0.546 \pm 0.04$ | $\mathbf{0.572 \pm 0.04}$ |
| Semantic E | $\mathbf{0.986 \pm 0.01}$ | $0.852 \pm 0.03$ | $0.398 \pm 0.04$ | $0.642 \pm 0.04$ | $0.460 \pm 0.04$ | $0.602 \pm 0.04$ | $0.506 \pm 0.04$ |

### (d) Subset

| Metric | Llama | Qwen-0.6B | Qwen-0.6B (R) | Qwen3-4B | Qwen-4B (R) | Mistral-7B | Qwen3-8B |
|---|---|---|---|---|---|---|---|
| Perplexity | $0.483 \pm 0.02$ | $0.374 \pm 0.02$ | $0.540 \pm 0.02$ | $0.477 \pm 0.02$ | $0.297 \pm 0.02$ | $0.450 \pm 0.02$ | $0.437 \pm 0.02$ |
| Energy | $0.501 \pm 0.02$ | $0.386 \pm 0.02$ | $0.467 \pm 0.02$ | $0.472 \pm 0.02$ | $0.266 \pm 0.02$ | $0.574 \pm 0.02$ | $0.352 \pm 0.02$ |
| Entropy | $0.448 \pm 0.02$ | $0.416 \pm 0.02$ | $0.474 \pm 0.02$ | $0.417 \pm 0.02$ | $0.478 \pm 0.02$ | $0.471 \pm 0.02$ | $0.432 \pm 0.02$ |
| Lex Sim | $0.706 \pm 0.02$ | $0.557 \pm 0.02$ | $\mathbf{0.751 \pm 0.02}$ | $0.547 \pm 0.02$ | $0.504 \pm 0.02$ | $0.688 \pm 0.02$ | $0.549 \pm 0.02$ |
| $E_{original}$ | $0.464 \pm 0.02$ | $0.522 \pm 0.02$ | $0.449 \pm 0.02$ | $0.456 \pm 0.02$ | $0.31 \pm 0.02$ | $0.512 \pm 0.02$ | $0.619 \pm 0.02$ |
| $E_{output}$ | $0.718 \pm 0.02$ | $\mathbf{0.684 \pm 0.02}$ | $0.744 \pm 0.02$ | $0.571 \pm 0.02$ | $\mathbf{0.613 \pm 0.02}$ | $0.771 \pm 0.02$ | $0.578 \pm 0.02$ |
| $E_{average}$ | $\mathbf{0.740 \pm 0.02}$ | $0.682 \pm 0.02$ | $0.727 \pm 0.02$ | $\mathbf{0.610 \pm 0.02}$ | $0.574 \pm 0.02$ | $\mathbf{0.779 \pm 0.02}$ | $\mathbf{0.709 \pm 0.02}$ |
| Semantic E | $0.504 \pm 0.02$ | $0.625 \pm 0.02$ | $0.641 \pm 0.02$ | $0.464 \pm 0.02$ | $0.605 \pm 0.02$ | $0.462 \pm 0.02$ | $0.490 \pm 0.02$ |

### (e) Union

| Metric | Llama | Qwen-0.6B | Qwen-0.6B (R) | Qwen3-4B | Qwen-4B (R) | Mistral-7B | Qwen3-8B |
|---|---|---|---|---|---|---|---|
| Perplexity | $0.533 \pm 0.04$ | $0.540 \pm 0.04$ | $0.426 \pm 0.04$ | $0.567 \pm 0.04$ | $0.437 \pm 0.06$ | $0.549 \pm 0.05$ | $0.584 \pm 0.04$ |
| Energy | $0.524 \pm 0.04$ | $0.550 \pm 0.04$ | $0.471 \pm 0.04$ | $0.563 \pm 0.05$ | $0.374 \pm 0.06$ | $0.530 \pm 0.05$ | $0.645 \pm 0.04$ |
| Entropy | $0.526 \pm 0.04$ | $0.480 \pm 0.04$ | $0.434 \pm 0.03$ | $0.566 \pm 0.04$ | $\mathbf{0.484 \pm 0.07}$ | $0.505 \pm 0.03$ | $0.550 \pm 0.04$ |
| Lex Sim | $0.585 \pm 0.05$ | $0.540 \pm 0.04$ | $0.356 \pm 0.05$ | $0.616 \pm 0.05$ | $0.363 \pm 0.06$ | $\mathbf{0.556 \pm 0.04}$ | $0.607 \pm 0.06$ |
| $E_{original}$ | $0.554 \pm 0.04$ | $0.525 \pm 0.04$ | $0.509 \pm 0.03$ | $0.568 \pm 0.04$ | $0.439 \pm 0.06$ | $0.504 \pm 0.04$ | $0.447 \pm 0.03$ |
| $E_{output}$ | $\mathbf{0.635 \pm 0.05}$ | $\mathbf{0.616 \pm 0.04}$ | $\mathbf{0.599 \pm 0.04}$ | $\mathbf{0.677 \pm 0.05}$ | $0.476 \pm 0.07$ | $0.506 \pm 0.04$ | $\mathbf{0.707 \pm 0.04}$ |
| $E_{average}$ | $0.569 \pm 0.05$ | $0.488 \pm 0.04$ | $0.431 \pm 0.04$ | $0.610 \pm 0.04$ | $0.460 \pm 0.07$ | $0.527 \pm 0.03$ | $0.586 \pm 0.05$ |
| Semantic E | $0.508 \pm 0.04$ | $0.477 \pm 0.03$ | $0.474 \pm 0.03$ | $0.529 \pm 0.04$ | $0.381 \pm 0.04$ | $0.477 \pm 0.03$ | $0.564 \pm 0.05$ |

### (f) Intersection

| Metric | Llama | Qwen-0.6B | Qwen-0.6B (R) | Qwen3-4B | Qwen-4B (R) | Mistral-7B | Qwen3-8B |
|---|---|---|---|---|---|---|---|
| Perplexity | $0.574 \pm 0.04$ | $0.476 \pm 0.04$ | $0.558 \pm 0.04$ | $0.477 \pm 0.04$ | $0.562 \pm 0.04$ | $0.412 \pm 0.04$ | $0.473 \pm 0.04$ |
| Energy | $0.578 \pm 0.04$ | $0.422 \pm 0.04$ | $0.464 \pm 0.04$ | $0.457 \pm 0.04$ | $0.469 \pm 0.04$ | $0.564 \pm 0.04$ | $0.461 \pm 0.04$ |
| Entropy | $0.615 \pm 0.04$ | $0.463 \pm 0.04$ | $0.548 \pm 0.04$ | $0.439 \pm 0.04$ | $0.475 \pm 0.05$ | $0.504 \pm 0.04$ | $0.500 \pm 0.04$ |
| Lex Sim | $0.646 \pm 0.04$ | $0.450 \pm 0.04$ | $0.645 \pm 0.04$ | $0.461 \pm 0.04$ | $0.587 \pm 0.04$ | $0.683 \pm 0.03$ | $0.494 \pm 0.03$ |
| $E_{original}$ | $0.473 \pm 0.04$ | $0.558 \pm 0.03$ | $0.562 \pm 0.03$ | $0.475 \pm 0.04$ | $0.541 \pm 0.04$ | $0.439 \pm 0.04$ | $0.538 \pm 0.03$ |
| $E_{output}$ | $0.596 \pm 0.05$ | $0.495 \pm 0.04$ | $\mathbf{0.728 \pm 0.03}$ | $0.452 \pm 0.04$ | $0.641 \pm 0.04$ | $0.655 \pm 0.04$ | $0.490 \pm 0.04$ |
| $E_{average}$ | $\mathbf{0.687 \pm 0.04}$ | $\mathbf{0.571 \pm 0.04}$ | $0.651 \pm 0.04$ | $0.505 \pm 0.04$ | $\mathbf{0.599 \pm 0.04}$ | $\mathbf{0.698 \pm 0.04}$ | $\mathbf{0.566 \pm 0.04}$ |
| Semantic E | $0.415 \pm 0.04$ | $0.503 \pm 0.04$ | $0.483 \pm 0.04$ | $\mathbf{0.524 \pm 0.04}$ | $0.439 \pm 0.04$ | $0.458 \pm 0.04$ | $0.463 \pm 0.04$ |

Table A16: Accuracy breakdown for each dataset and for each model (without excluding low-quality responses).

### (a) **Complement**

| Metric | Llama | Qwen-0.6B | Qwen-4B | Mistral-7B | Qwen-8B |
|---|---|---|---|---|---|
| Perplexity | $0.609 \pm 0.09$ | $0.412 \pm 0.1$ | $0.485 \pm 0.09$ | $0.200 \pm 0.2$ | $0.443 \pm 0.09$ |
| Energy | $0.700 \pm 0.09$ | $0.353 \pm 0.1$ | $0.485 \pm 0.09$ | $0.700 \pm 0.3$ | $0.214 \pm 0.07$ |
| Entropy | $0.727 \pm 0.08$ | $0.368 \pm 0.1$ | $0.697 \pm 0.08$ | $0.200 \pm 0.2$ | $0.351 \pm 0.08$ |
| Lex Sim | $0.900 \pm 0.06$ | $0.500 \pm 0.1$ | $0.674 \pm 0.08$ | $0.100 \pm 0.2$ | $0.527 \pm 0.09$ |
| $E_{\text{original}}$ | $0.554 \pm 0.09$ | $0.691 \pm 0.1$ | $0.667 \pm 0.08$ | $0.800 \pm 0.2$ | $0.550 \pm 0.09$ |
| $E_{\text{output}}$ | $\mathbf{0.946} \pm 0.04$ | $\mathbf{0.956} \pm 0.05$ | $\mathbf{0.864} \pm 0.06$ | $0.700 \pm 0.3$ | $\mathbf{0.756} \pm 0.07$ |
| $E_{\text{average}}$ | $0.927 \pm 0.05$ | $0.672 \pm 0.11$ | $0.589 \pm 0.11$ | $\mathbf{0.900} \pm 0.2$ | $0.621 \pm 0.11$ |
| Semantic E | $0.391 \pm 0.09$ | $0.691 \pm 0.1$ | $0.386 \pm 0.08$ | $0.500 \pm 0.3$ | $0.267 \pm 0.08$ |

### (b) **SyntheticQA**

| Metric | Llama | Qwen-0.6B | Qwen-4B | Mistral-7B | Qwen-8B |
|---|---|---|---|---|---|
| Perplexity | $0.643 \pm 0.05$ | $0.658 \pm 0.11$ | $0.290 \pm 0.06$ | $0.049 \pm 0.04$ | $0.318 \pm 0.05$ |
| Energy | $0.678 \pm 0.05$ | $0.608 \pm 0.11$ | $0.383 \pm 0.06$ | $0.382 \pm 0.09$ | $0.365 \pm 0.05$ |
| Entropy | $0.640 \pm 0.05$ | $0.506 \pm 0.11$ | $0.262 \pm 0.06$ | $0.402 \pm 0.10$ | $0.456 \pm 0.05$ |
| Lex Sim | $0.490 \pm 0.05$ | $0.683 \pm 0.10$ | $0.278 \pm 0.06$ | $\mathbf{0.578} \pm 0.10$ | $0.236 \pm 0.05$ |
| $E_{\text{original}}$ | $0.461 \pm 0.05$ | $0.418 \pm 0.11$ | $0.222 \pm 0.05$ | $0.402 \pm 0.10$ | $0.368 \pm 0.05$ |
| $E_{\text{output}}$ | $0.714 \pm 0.05$ | $\mathbf{0.861} \pm 0.08$ | $0.371 \pm 0.06$ | $0.392 \pm 0.10$ | $0.358 \pm 0.05$ |
| $E_{\text{average}}$ | $\mathbf{0.793} \pm 0.04$ | $0.672 \pm 0.11$ | $\mathbf{0.589} \pm 0.11$ | $0.529 \pm 0.10$ | $\mathbf{0.621} \pm 0.11$ |
| Semantic E | $0.349 \pm 0.05$ | $0.354 \pm 0.11$ | $0.298 \pm 0.06$ | $0.500 \pm 0.10$ | $0.355 \pm 0.05$ |

### (c) **Random Choice**

| Metric | Llama | Qwen-0.6B | Qwen-4B | Mistral-7B | Qwen-8B |
|---|---|---|---|---|---|
| Perplexity | $0.673 \pm 0.04$ | $0.504 \pm 0.05$ | $0.679 \pm 0.04$ | $0.458 \pm 0.2$ | $0.456 \pm 0.04$ |
| Energy | $0.608 \pm 0.04$ | $0.697 \pm 0.04$ | $\mathbf{0.846} \pm 0.03$ | $0.667 \pm 0.2$ | $0.314 \pm 0.04$ |
| Entropy | $0.827 \pm 0.03$ | $0.391 \pm 0.05$ | $0.661 \pm 0.04$ | $0.583 \pm 0.2$ | $0.471 \pm 0.04$ |
| Lex Sim | $0.610 \pm 0.04$ | $0.740 \pm 0.04$ | $0.643 \pm 0.04$ | $0.708 \pm 0.2$ | $0.471 \pm 0.04$ |
| $E_{\text{original}}$ | $0.785 \pm 0.04$ | $0.733 \pm 0.04$ | $0.575 \pm 0.05$ | $0.708 \pm 0.2$ | $0.544 \pm 0.04$ |
| $E_{\text{output}}$ | $0.683 \pm 0.04$ | $\mathbf{0.853} \pm 0.03$ | $0.675 \pm 0.04$ | $0.667 \pm 0.2$ | $0.564 \pm 0.04$ |
| $E_{\text{average}}$ | $0.477 \pm 0.05$ | $0.672 \pm 0.11$ | $0.589 \pm 0.11$ | $\mathbf{0.792} \pm 0.2$ | $\mathbf{0.621} \pm 0.11$ |
| Semantic E | $\mathbf{0.988} \pm 0.01$ | $0.844 \pm 0.03$ | $0.600 \pm 0.05$ | $0.417 \pm 0.2$ | $0.507 \pm 0.04$ |

### (d) **Subset**

| Metric | Llama | Qwen-0.6B | Qwen-4B | Mistral-7B | Qwen-8B |
|---|---|---|---|---|---|
| Perplexity | $0.403 \pm 0.03$ | $0.357 \pm 0.03$ | $0.438 \pm 0.03$ | $0.455 \pm 0.05$ | $0.394 \pm 0.03$ |
| Energy | $0.426 \pm 0.03$ | $0.386 \pm 0.03$ | $0.460 \pm 0.03$ | $0.617 \pm 0.05$ | $0.331 \pm 0.02$ |
| Entropy | $0.371 \pm 0.03$ | $0.390 \pm 0.03$ | $0.407 \pm 0.03$ | $0.513 \pm 0.05$ | $0.367 \pm 0.02$ |
| Lex Sim | $0.730 \pm 0.03$ | $0.614 \pm 0.03$ | $0.541 \pm 0.03$ | $0.751 \pm 0.05$ | $0.561 \pm 0.03$ |
| $E_{\text{original}}$ | $0.517 \pm 0.03$ | $0.562 \pm 0.03$ | $0.473 \pm 0.03$ | $0.513 \pm 0.05$ | $0.599 \pm 0.03$ |
| $E_{\text{output}}$ | $0.743 \pm 0.03$ | $\mathbf{0.756} \pm 0.03$ | $0.585 \pm 0.03$ | $0.823 \pm 0.04$ | $0.614 \pm 0.03$ |
| $E_{\text{average}}$ | $\mathbf{0.779} \pm 0.03$ | $0.672 \pm 0.11$ | $\mathbf{0.589} \pm 0.11$ | $\mathbf{0.852} \pm 0.04$ | $\mathbf{0.621} \pm 0.11$ |
| Semantic E | $0.547 \pm 0.03$ | $0.590 \pm 0.03$ | $0.499 \pm 0.03$ | $0.499 \pm 0.05$ | $0.500 \pm 0.03$ |

### (e) **Union**

| Metric | Llama | Qwen-0.6B | Qwen-4B | Mistral-7B | Qwen-8B |
|---|---|---|---|---|---|
| Perplexity | $0.534 \pm 0.04$ | $0.533 \pm 0.07$ | $0.570 \pm 0.04$ | $\mathbf{0.545} \pm 0.07$ | $0.580 \pm 0.05$ |
| Energy | $0.528 \pm 0.04$ | $0.544 \pm 0.05$ | $0.569 \pm 0.05$ | $0.542 \pm 0.07$ | $0.642 \pm 0.04$ |
| Entropy | $0.532 \pm 0.04$ | $0.491 \pm 0.05$ | $0.563 \pm 0.05$ | $0.469 \pm 0.07$ | $0.552 \pm 0.04$ |
| Lex Sim | $0.580 \pm 0.05$ | $0.565 \pm 0.06$ | $0.608 \pm 0.05$ | $0.517 \pm 0.07$ | $0.604 \pm 0.06$ |
| $E_{\text{original}}$ | $0.552 \pm 0.04$ | $0.509 \pm 0.06$ | $0.568 \pm 0.04$ | $0.511 \pm 0.06$ | $0.448 \pm 0.04$ |
| $E_{\text{output}}$ | $\mathbf{0.636} \pm 0.05$ | $0.623 \pm 0.06$ | $\mathbf{0.673} \pm 0.05$ | $0.483 \pm 0.08$ | $\mathbf{0.708} \pm 0.04$ |
| $E_{\text{average}}$ | $0.572 \pm 0.05$ | $\mathbf{0.672} \pm 0.11$ | $0.589 \pm 0.11$ | $0.538 \pm 0.06$ | $0.621 \pm 0.11$ |
| Semantic E | $0.505 \pm 0.04$ | $0.403 \pm 0.06$ | $0.525 \pm 0.04$ | $0.497 \pm 0.06$ | $0.565 \pm 0.05$ |

### (f) **Intersection**

| Metric | Llama | Qwen-0.6B | Qwen-4B | Mistral-7B | Qwen-8B |
|---|---|---|---|---|---|
| Perplexity | $0.564 \pm 0.04$ | $0.490 \pm 0.05$ | $0.483 \pm 0.04$ | $0.450 \pm 0.09$ | $0.475 \pm 0.04$ |
| Energy | $0.573 \pm 0.04$ | $0.413 \pm 0.06$ | $0.464 \pm 0.04$ | $0.550 \pm 0.08$ | $0.464 \pm 0.04$ |
| Entropy | $0.626 \pm 0.04$ | $0.486 \pm 0.05$ | $0.443 \pm 0.04$ | $0.557 \pm 0.08$ | $0.491 \pm 0.04$ |
| Lex Sim | $0.642 \pm 0.04$ | $0.436 \pm 0.05$ | $0.465 \pm 0.04$ | $0.669 \pm 0.09$ | $0.498 \pm 0.03$ |
| $E_{\text{original}}$ | $0.468 \pm 0.04$ | $\mathbf{0.601} \pm 0.04$ | $0.479 \pm 0.04$ | $0.601 \pm 0.08$ | $0.547 \pm 0.03$ |
| $E_{\text{output}}$ | $0.600 \pm 0.05$ | $0.515 \pm 0.05$ | $0.453 \pm 0.04$ | $0.660 \pm 0.07$ | $0.498 \pm 0.04$ |
| $E_{\text{average}}$ | $\mathbf{0.682} \pm 0.04$ | $0.598 \pm 0.06$ | $\mathbf{0.497} \pm 0.06$ | $\mathbf{0.678} \pm 0.07$ | $\mathbf{0.577} \pm 0.04$ |
| Semantic E | $0.426 \pm 0.04$ | $0.496 \pm 0.04$ | $0.426 \pm 0.04$ | $0.459 \pm 0.09$ | $0.473 \pm 0.04$ |

