# OpenReview forum: "Generation Space Size:  Understanding and Calibrating Open-Endedness of LLM Generations"
_ICLR.cc/2026/Conference — Submitted to ICLR 2026_

### Official Review · Reviewer_KJg8 · 2025-10-28

**Soundness:** 3
**Presentation:** 3
**Contribution:** 4
**Rating:** 6
**Confidence:** 3

**Summary:**

The paper “Generation Space Size: Understanding and Calibrating Open-Endedness of LLM Generations” introduces the concept of Effective Generation Space Size (GSS) — the number of semantically distinct outputs a large language model (LLM) considers for a given prompt. The authors argue that many generation failures (e.g., overly homogeneous outputs in creative tasks or excessive hallucination in factual tasks) stem from miscalibration of this generation space: models either over-expand or over-collapse their internal search space.

To study this phenomenon, the paper proposes GSSBench, a benchmark consisting of prompt pairs with known ground-truth GSS relationships. Using this framework, the authors systematically compare various metrics (diversity, uncertainty, and hallucination-based measures) to estimate a model’s GSS. They find that hallucination detection metrics—especially EigenScore—consistently approximate true GSS better than traditional diversity or uncertainty measures and provide interpretable insights into model behavior.

**Strengths:**

1. Defines Effective Generation Space Size (GSS) as a unifying measure of output open-endedness in LLMs.

2. Methodological: Introduces GSSBench, the first benchmark for evaluating GSS calibration across tasks.

**Weaknesses:**

While the paper claims that GSS estimation can improve hallucination detection and creative text generation, it does not clearly quantify how much GSS-based calibration outperforms existing baselines. The experiments mainly focus on correlation analyses rather than downstream task gains. Demonstrating concrete improvements (e.g., reduction in hallucination rate or increase in creativity scores) would strengthen the argument that GSS provides practical benefits beyond theoretical interpretability.

**Questions:**

as weakness

---

> ### Author Response · Authors · 2025-11-21
>
> Thank you so much for reading our paper, and for the thoughtful feedback!
>
> **Re: experiments mainly focus on correlation analyses**:
> Thank you for this point! [1] showed that a test time feature clipping method can reduce overconfident generations. We showed that in addition to this existing application, the same method can be used to increase diversity on open-ended tasks. Additionally, in an experiment in Appendix C3, we show that the metrics have comparable and potentially better performance than GPT-4o at the grounding-act classification task, which [2] has identified as extremely difficult. We also show that these metrics provide helpful signals of prompt ambiguity and when an LLM would ask a clarification question in a sample of responses. Future work can directly explore the optimal threshold of when an LLM should ask for clarification instead of directly providing a generation and directly intervening to better control a model’s grounding behavior. Given that grounding failure is a subjective human-AI interaction challenge and no existing benchmark can directly quantify improvement, we encourage future work to leverage our findings to explore solutions.
>
> You are right to put out that most of our analyses are correlational and not focused on benchmark improvement. We believe that having a quantification of GSS is an important first step towards improving LLMs’ performance on various use cases. We have updated the limitation section and edited the paper throughout to make this point clearer.
> We would like to thank you again for reading and engaging with our paper!
>
>
> [1] Chen, C., Liu, K., Chen, Z., Gu, Y., Wu, Y., Tao, M., ... & Ye, J. (2024). INSIDE: LLMs' internal states retain the power of hallucination detection. arXiv preprint arXiv:2402.03744.
>
> [2] Shaikh, O., Mozannar, H., Bansal, G., Fourney, A., & Horvitz, E. (2025). Navigating rifts in human-llm grounding: Study and benchmark. arXiv preprint arXiv:2503.13975.

---

### Official Review · Reviewer_W64U · 2025-10-31

**Soundness:** 1
**Presentation:** 2
**Contribution:** 2
**Rating:** 2
**Confidence:** 3

**Summary:**

This paper considers an intuitive setup to the issue of model output diversity —  the space valid generation candidates (and its size — Generation Space Size, aka GSS). The author proposes GSS mismatch, where the lack of diversity is covering lower GSS than allowed, and hallucination, too high.
To that end, the paper proposes GSS bench, where prompt pairs with relative GSS relationship known available for comparison, and various assess the best proxy of model GSS coverage.

**Strengths:**

- Using multiple datasets, this works made a good effort to cover various topics for measuring model GSS coverage.
- Compared many existing signals from lexical level to semantic one as a proxy for GSS coverage, including improved variants.
- Studied GSS’s correlation with various effects like question ambiguity, reasoning lengthen, solution paths, and value as a loss.

**Weaknesses:**

Unfortunately, this paper has some fundamental issues
- No explicit mention of any utility evaluation of generated candidates. For all we know, models can be generating non fluent text and still be considered high GSS.
- The paper assumes prompt-pair orderings can predict GSS. However, such automated construction might only reveal surface-form notion of distinct outputs. So the generalizability is of concern - investigations of these multiple metrics might be fitting for assumptions more than actually GSS.
- The paper says it is “currently impossible to access the model’s generation space … unless we sample infinitely many times” but then proceeds to estimate all metrics from K=10 samples. Yet, the paper itself assumes one can work with small-K samples. Hence, the motivation to find proxy metrics seems conflicted. I
- It is not informative to simply assumes both error terms are “small enough” so metric orderings transfer. Nowhere do we see, for at least one dataset/model, the rate of violations of these assumptions, which can be a reliability test for GSS Bench.
- Some claims are post-hoc. To “predict” clarifications or ambiguity or #solution paths, the method has already sampled multiple responses from the model — at that point you can already see whether the model asked for clarification.
- EigenScore, using more samples, is more costly than other metrics. Hence all comparison were made fair.

**Questions:**

- Please edit “all reported values have ±0.02 margin of error”
- I realized you used Llama 3.1 only from the citation. Please actually say so in the paper instead of just saying "Llama 8B".

---

> ### Author Response · Authors · 2025-11-21
>
> Thank you so much for reading and engaging with our paper! We hope our responses below can address your concerns.
>
> **Re: no explicit mention of utility**:
> Thank you for this point. Indeed GSS does not capture utility (which we explicitly acknowledge as a limitation), and instead should be used in tandem with other utility evaluations such as a reward model. Instead it captures a new concept of generation space size. Like other related work on improving diversity of outputs or reducing hallucinations,  we are interested in studying GSS of  models that do already produce fluent text: All five models we used are instruction-tuned and can respond to prompts appropriately. To empirically validate this, we have added an LLM judge (GPT-4o) and rated the validity of each response. We found that across all models, the rate of valid responses is 69.4% (80% for Llama-8B-Instruct, 35% for Mistral, 63% for Qwen-0.6B, 83% for Qwen-4B, and 87% for Qwen-8B). We have excluded the low-quality responses in our accuracy calculation and updated the results in Table 2. The main findings remain the same: E-Output and E-Average are still the best metrics.
>
> **Re: surface-level differences**:
> Uncertainty quantification metrics and diversity metrics both aim to capture the notion of distinct outputs, but there is no systematic way to compare them. GSSBench fills this gap by directly allowing us to compare what each metric actually tracks for the same model. To validate the construction of the prompt pairs, we used an external model GPT-4o to determine which prompt in a pair warrants a bigger generation space. We find that there is a high agreement between GPT judgments and our ground-truth prompt construction. The results can be found in Table A9. For all datasets except for intersection, there is almost perfect agreement, confirming the validity of our choices.
>
> **Re: small sample size K**:
> We wrote “currently impossible to access the model’s generation space… unless we sample infinitely times” to point out that GSS is a non-interpretable concept that is difficult to probe. Therefore, we introduce GSSBench as a framework to explore efficient ways to approximate it using fewer samples. We explore ablations of the sample size K (see Appendix B, Figure A6) and found that across different datasets, the performance (i.e. accuracy on GSSBench) stopped improving after the sample size reaches 20, and the improvement from K=10 to 20 is minimal, which matches with sample size ablations in [1]. Therefore, using a smaller sample size and higher temperature is sufficient to approximate a model’s GSS. To fully address your concern, we use the number of unique, valid outputs on the Random Choice dataset as the proxy of a model’s actual generation space size and find that the value remains stable despite the increase in K up to K = 50. Therefore, the motivation to use smaller samples as proxies is fully justified and is the common practice for diversity evaluations. A smaller sample size and a model’s representation of this space can already reveal much about a model’s miscalibration, in the same way that a small sample size can help detect a model’s uncertainty about a factualQA question. We have added the plot Figure A8.
>
> **Re: assumptions about error terms**:
> Thank you for this point. To show that this is a reasonable assumption, on the Random Choice dataset, the only dataset where the model output space is readily verifiable, we calculated the number of distinct responses across different sample sizes and report below the rate of violations, i.e., number of cases where the model behaves not as we expect (smaller generation space empirically for the prompt we expect), and see that this number is reasonably small:
> Llama: 0.7%
> Mistral: 4.4%
> Qwen-0.6B: 1.1%
> Qwen-4B: 7.5%
> Qwen-8B: 5%
> Excluding these cases of violations, we recalculated the accuracy and found that when we eliminate the model miscalibration error, the metric orderings still transfer, i.e. the same metric that had the best performance still has the highest accuracy for the same model. We have updated this result in Table A12.

---

> > ### Author Response · Authors · 2025-11-21
> >
> > **Re: claims are post-hoc**:
> > Our claims are analogous to claims that hallucination detection metrics can predict when models hallucinate, which also requires sampling multiple responses from the model. Our experiments showed that without sampling, uncertainty quantification methods (e.g. perplexity and entropy) alone cannot distinguish between ambiguous prompts from non-ambiguous prompts, similar to how these traditional metrics cannot predict if a model hallucinates for a particular question. Additionally, our finding that EigenScore Output has optimal performance shows that instead of using a model’s hidden states, we can simply use an external and much smaller model to generate a score that is informative of whether the original model would end up asking a clarification question. The finding provides insights that when models ask a clarification question, they inherently represent the generation space as large, resulting in samples that are semantically different.
> >
> > **Re: computational cost**:
> > This is a good point. However, note that EigenScore Output is much cheaper than the original EigenScore implementation from [1] because instead of taking the hidden states across each sample, it takes the embedding of each generation generated by a sentence transformer, which doesn’t require storing the hidden states of the original model. We also show that cheaper methods like lexical similarity fail to actually capture a model’s GSS, even when sample size is increased to 50.
> >
> > We would like to thank you again for reading and engaging with our paper, and for the thoughtful suggestion!
> >
> > [1] Chen, C., Liu, K., Chen, Z., Gu, Y., Wu, Y., Tao, M., ... & Ye, J. (2024). INSIDE: LLMs' internal states retain the power of hallucination detection. arXiv preprint arXiv:2402.03744.

---

### Official Review · Reviewer_xAGv · 2025-11-01

**Soundness:** 2
**Presentation:** 3
**Contribution:** 2
**Rating:** 4
**Confidence:** 4

**Summary:**

The paper hypothesizes that the homogenization of outputs in creative tasks and the occurrence of hallucinations in factual tasks both arise from a miscalibration of the Generation Space Size in LMs. Since direct quantification of this space is tricky, the authors propose, GSSBench, to assess how well existing metrics approximate GSS. They identify EigenScore as the most effective proxy and demonstrate some applications of GSS including detecting prompt ambiguity and analyzing divergence from optimal reasoning token budgets.

**Strengths:**

- The paper has some interesting insights that are corroborated by literature in diverse generations; For instance, the paper demonstrates that larger models show higher miscalibration in GSS from a different angle (something that’s already anecdotally reported in literature already).

**Weaknesses:**

- The use of Eigen (Avg + Output) for calibrating the GSS for a wider variety of prompts where GSS might be conflated or constrained due to other reasons is unclear:

I am not entirely convinced about the soundness of the design decision that the output space of X is more constrained than Y (agnostic to the type of constraint that's introduced). For the motivating example -  “Generate an email that contains the word Sam” having a smaller GSS  than “Generate an email” seems plausible but is seems strongly conditioned on the inclusion of a proper noun constraint. But if the same constraint is replaced by ‘climate’ - the generation space may or may not become constrained depending upon the topical associations seen in pretraining/post-training for such a constraint. Empirically speaking - For Figure 2: the small proportion of overlapping buckets between complements and originals is not addressed - perhaps such prompts elicit such overlapping score ranges and why ?

Figure A2 also shows that Eigen Variant has complete overlap with the original distribution for the Random Choice dataset so it’s further unclear which of the 3 Eigen variants are ultimately useful.


- GSS reasoning-length correlation is weak at best: Ln 334 (Experiment 2) claims that GSS is predictive of reasoning complexity. Figure 3 demonstrates this weakly. The consistent winner for the previously winning method E-Average shows a maximum correlation of 0.17 and several of the sizes/datasets show negative to no correlation. It also seems a lot of the other methods show even higher correlation with the reasoning length so it's unclear if the takeaway is supposed to be that EigenScore is independently useful for such prediction or the entire class of metrics can mostly do the job.

- In general, picking a single method among the EigenScore variants is also recommended since keeping a track of which of the variant’s outperforms is cumbersome.

**Questions:**

- The claim that GSS measures prompt ambiguity is insufficiently validated. The results demonstrate population-level separation but not instance-level classification. Why not report metrics such as F1 or ROC-AUC to quantify per-sample discriminability?
- Ln 161: Missing space.
- Figure 2: Font size is too small to be legible.
- In the LOOE experiment (Section 4.3), could the authors clarify how the threshold parameter p (e.g., p = 0.6 in Table 5) was selected and whether this choice was validated on a held-out set or tuned on the test data? This detail is essential for assessing the robustness of the reported improvements.

---

> ### Author Response · Authors · 2025-11-21
>
> Thank you for the detailed and thoughtful review!
>
> **Re “design decision that the output space of X is more constrained than Y”**:
>
> Thank you for pointing this out! We would like to clarify the distinction between a model’s GSS and the theoretical GSS. What you suggested – that model generations are highly conditioned on the appended contents (and their frequency in training data) – is true, and is in fact the kind of miscalibration that we hope to capture using GSSBench.
> To validate the construction of the prompt pairs, we used an external model GPT-4o to determine which prompt in a pair warrants a bigger generation space. We find that there is a high agreement between GPT judgments and our ground-truth prompt construction. The results can be found in Table A9. For all datasets except for intersection, there is almost perfect agreement, confirming the validity of our choices.
>
> **Re “small proportion of overlapping buckets between complements and originals”**:
> We do not know the absolute generation size for each prompt, but what our evaluation framework enables is the pairwise comparison between the GSS for across each prompt pair. In comparing GSS of two prompts, we have a relational way of talking about the GSSs of the model, given the two prompts. Moreover, the relational way of talking about GSS in fact emphasizes the virtue of the GSSbench design in that it makes it possible to talk about GSS more accurately.  Figure 2 shows the distribution of values across the “original” prompts and the “complement” versions, where we expect that, in general, a model’s GSS for original prompts should be relatively smaller than the complement versions. The overlap reveals places where a model’s GSS for a complement prompt is relatively smaller: for example, some complement prompts with very small GSS include “Generate anything that is not a Python program about sorting a list with recursion that uses type hints and is modularized into separate files” and “Generate anything that is not an email about job opportunities in the non-profit sector that mentions my attached résumé and follows the outline: 1) Greeting 2) Purpose 3) Qualifications 4) Next steps.” For the second prompt, instead of generating emails, models often generate a “letter about job opportunities…”, and each of the ten samples are extremely homogenous. Your observation is indeed an interesting behavior of LLMs, and we’ve added examples of prompts in the overlapping area in Table A14.
> **Re "Figure A2 also shows that Eigen Variant has complete overlap with the original distribution for the Random Choice dataset so it’s further unclear which of the 3 Eigen variants are ultimately useful."
> Thanks for pointing this out! We introduce different dataset types (e.g. Random Choice, Complement, etc.) to encompass different types of open-ended generations. We include the accuracy per dataset in Table A10. From the table, we observe that for all tasks except for Random Choice, EigenScore Output and EigenScore Variant are consistently better-performing. Your observation about A2 is spot-on: the EigenScore metrics indeed have a lower accuracy score on this particular task for different models, likely because the output tokens tend to be shorter. We have updated our limitation section to reflect this.
>
> **Re: reasoning token length**:
> Thank you for pointing this out! We have adjusted this section to highlight that reasoning token length (which [1] have found to predict human reaction times and task difficulty) has a positive correlation with most of the metrics we test and added an additional analysis showing that positive correlations only exist for strictly deductive tasks and are negative for controversial logic tasks, as noted in [2]. The takeaway is that most metrics are positively correlated with reasoning length on deductive tasks, but this positive correlation may not hold for other reasoning tasks, where the relationship between GSS and a reasoning task’s complexity can be more complicated.
>
> **Re: picking one method for EigenScore**:
> Thank you for the suggestion. We have removed EigenScore Original (the implementation used in [3]). We are keeping EigenScore Output and EigenScore Variant because EigenScore Output is much more light-weight, and their equally good performance shows that using a model’s own hidden state representations is not required to approximate a model’s GSS.

---

> ### Author Response · Authors · 2025-11-21
>
> **Re: Q1**:
> The finding that EigenScore Output is the only metric among all candidates with the ability to separate ambiguous prompts from non-ambiguous ones further corroborates that EigenScore Output encodes useful information about a model’s prompt representation. We included an analysis of instance-level classification (see Appendix C3, Table A13). We used threshold classifiers using each metric and reported the Macro-F1 and AUC, compared to the baseline of prompting GPT-4o. We found that most metrics yield above-chance performance and are comparable to directly prompting a language model. Both experiments show that GSS metrics capture prompt ambiguity more broadly.
>
> **Re: Q2, Q3**:
> Thank you for these suggestions. We have adjusted the font size of the figure.
>
> **Re: Q4**:
> For each diversity metric (NLL, LOOE, and Lex Sem), we ran experiments using 6 different p-values: 0.1, 0.2, 0.3, 0.4, 0.5, and 0.6, and the full results are included in Table A24. For Table 5, we report the p-value with diversity increase without a decrease in quality (as scored through the reward model) compared to the baseline model.
>
> Thank you again for the thoughtful review! We hope our responses were able to address most of your concerns.
>
> [1] de Varda, A. G., D’Elia, F. P., Lampinen, A., & Fedorenko, E. (2025). The cost of thinking is similar between large reasoning models and humans. ArXiV Preprint.
>
> [2] Holliday, W. H., Mandelkern, M., & Zhang, C. E. (2024). Conditional and modal reasoning in large language models. arXiv preprint arXiv:2401.17169.
>
> [3] Chen, C., Liu, K., Chen, Z., Gu, Y., Wu, Y., Tao, M., ... & Ye, J. (2024). INSIDE: LLMs' internal states retain the power of hallucination detection. arXiv preprint arXiv:2402.03744.

---

### Author Response · Authors · 2025-11-30

We thank the reviewers for their thoughtful comments and suggestions, and we are grateful that they find the work novel and interesting. We’ve responded to each review separately and believe that we have addressed all reviewer concerns with clarifications and additional experiments. In summary, our main additions are:

1. Validation of the ground-truth prompt pairs: to validate that our prompt pairs reflect a difference in the true generation space size, we prompted GPT-4o to identify the prompt with the bigger generation space size in each pair and found high agreement. The results are added to Table A9.

2. Response validity: to address the concern that there is no explicit mention of utility, we used GPT-4o to filter out all invalid responses and updated the accuracy scores to only include instances when responses are valid. We found that the metric orderings remain the same. We have updated the results in Table 2 and added more details in the Appendix.

3. Sample size validation: to address the concern that a sample size of 10 is too small to approximate a model’s GSS, we added an additional experiment showing that the number of unique responses do not increase significantly on the Random Choice dataset when sample size increases, justifying our current choice, which we provide further ablation studies for in the Appendix.

4. Error term robustness check: to provide a robustness check of the rate of violations of the assumptions that the error terms are small enough that metric orderings transfer, we directly calculate the rate of violations on the Random Choice dataset, where we operationalize a model’s GSS on the task as the number of distinct outputs across samples and report the proportion of times when the number is greater for the prompt with fewer options to choose from. We find that the rates are reasonably small across all models (ranging from 0.7% to 7.5%), and we re-calculated the accuracy and found that metric orderings still transfer. The full results can be found in Table A12.

5. Reasoning token length experiment adjustment: In light of the confusion around the takeaway of the reasoning token length experiment, we edited the section to highlight that positive correlations only exist for deductive reasoning tasks.

---

### Meta-Review · Area_Chair_pqra · 2026-01-09

**Summary:**

This paper analyzed two failure modes in open-ended generation tasks and proposed to new metric GSS to address those issue by measuring  the semantic misalignment between input and output along with a new benchmark GSSBench to assess model behaviours. While the merits of the paper (e.g.,  novel and interesting) appreciated by reviewers, there are several key weaknesses that concern the reviewers (and myself). Although the authors partially addressed some of these concerns during the rebuttal, major weaknesses remain.

**Reviewer Concerns:**

In particular, after a quick examination of the paper and a careful review of all rebuttals, I found that I share the same concerns raised by the reviewers, as summarized below:
- One major concern echoed by the reviewers is the utility of proposed evaluation metric and the soundness of the design decision. I agree with reviewer *W64U*'s concern of *models can be generating non fluent text and still be considered high GSS.* While the authors partially addressed this issue by adding an LLM-as-a-judge audit in the rebuttal, human evaluation would be more convincing (this also works for the concerns of “Validation of the ground-truth prompt pairs”).
- There are other concerns raised by reviewers, including lmotivation of use small sample size;  error term robustness check, and missing comparison with existing baselines. While the authors have partially addressed these points in the rebuttal (e.g., increasing sample size to 50, additional experiments on the grounding-act classification task), I encourage them to incorporate these analyses into the paper in subsequent revisions.

Addressing the issues outlined above would make the paper impactful and I encourage the authors to integrate these additional content into the next version of the paper as it will strengthen their contribution more.

**Reviewer Scores:**

I do not think the reviewers would have changed their scores as their major concerns mentioned above (also of mine) still remain.

---

### Decision · Program_Chairs · 2026-01-26

Reject